# DP-KFC: Data-Free Preconditioning for Privacy-Preserving Deep Learning

**Marc Molina Van den Bosch** [1 2]   **Riccardo Taiello** [1]   **Albert Sund Aillet** [1]   **Andrea Protani** [1 2]
**Miguel Angel Gonzalez Ballester** [2 3]   **Luigi Serio** [1]

## Abstract

Differentially private optimization suffers from a fundamental geometric mismatch: deep networks have highly anisotropic loss landscapes, yet DP-SGD injects isotropic noise. Second-order preconditioning can resolve this, but estimating curvature typically requires private data (consuming privacy budget) or public data (introducing distribution shift). We show that the Fisher Information Matrix decouples into *architectural sensitivity*, recoverable via synthetic noise, and *input correlations*, approximable from modality-specific frequency statistics. We propose DP-KFC, which constructs KFAC preconditioners by probing networks with structured synthetic noise, requiring neither private nor public data. Empirically, DP-KFC consistently outperforms DP-SGD and adaptive baselines across diverse modalities in strong privacy regimes ($\varepsilon \leq 3$). DP-KFC matches private-data preconditioners while public-data variants degrade by up to $4.8\%$, showing that curvature can be estimated without consuming privacy budget or introducing distribution shift. This enables privacy-preserving learning in specialized domains (e.g., medical applications) where regulatory constraints make data scarce.

## 1. Introduction

Differential privacy has emerged as the gold standard for protecting sensitive data in machine learning, yet its practical adoption remains limited by a fundamental geometric conflict (Abadi et al., 2016). Modern neural networks exhibit highly ill-conditioned loss landscapes where eigenvalues of the curvature matrix vary by orders of magnitude across parameter directions (Dauphin et al., 2014). Meanwhile, DP-

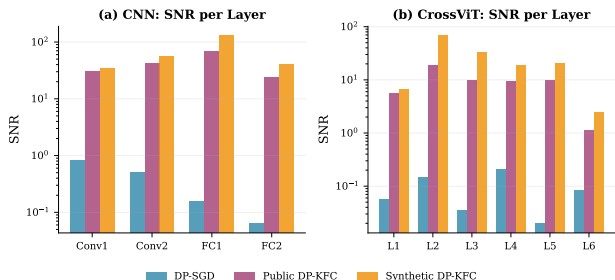

*Figure 1.* **Architectural Preconditioning.** Comparison of Layer-wise Signal-to-Noise Ratio (SNR) on (a) Simple CNN and (b) CrossViT-240 (Transformer) trained on CIFAR-100. Standard DP-SGD (Blue) suffers from signal collapse. DP-KFC reaches an optimal SNR profile across both local convolution and global attention architectures, matching the geometry of data-dependent proxies (Green) without accessing external data.

SGD enforces spherical constraints via global $L_2$ clipping and isotropic noise injection (Dwork & Roth, 2013), treating all directions equally. This *isotropy mismatch* causes low-sensitivity parameters to drown in noise while high-sensitivity directions are over-clipped (Chen et al., 2020), creating heterogeneous signal-to-noise ratios (SNR) (Fig. 1) that slow convergence and degrade performance (Tramèr & Boneh, 2021). Unlike non-private regimes where over-parameterization accelerates learning, private noise scales with model dimensionality ($\sqrt{d}$) (Bassily et al., 2014), often preventing deep networks from outperforming simple baselines.

Resolving this conflict via adaptive optimization has proven suboptimal. Standard adaptive methods (Kingma & Ba, 2017; Martens & Grosse, 2015) attempt to estimate second-moment statistics from the training data. However, as recent work by Ganesh et al. (2025) demonstrates, aiming for unbiased estimates of these statistics in a private setting is fundamentally misguided; the noise required to privatize the preconditioner often destabilizes the update more than the preconditioning helps. Similarly, Krouka et al. (2025) note that while second-order information is vital for convergence, in federated setups communicating or computing it privately introduces prohibitive overheads. Consequently, the field has pivoted toward *Precondition-then-Privatize* strategies that rely on public data to approximate the geometry (Amid et al., 2022; Li et al., 2022). Yet, this reliance is fragile: it

[1]CERN, Geneva, Switzerland [2]BCN MedTech, Department of Engineering, Universitat Pompeu Fabra, Barcelona, Spain [3]ICREA, Barcelona, Spain. Correspondence to: Marc Molina Van den Bosch <marc.molina.van.den.bosch@cern.ch>.

*Proceedings of the $43^{rd}$ International Conference on Machine Learning*, Seoul, South Korea. PMLR 306, 2026. Copyright 2026 by the author(s).

introduces unquantified biases, fails in specialized domains where proxies do not exist (e.g., medical imaging), and complicates privacy auditing (Tramèr et al., 2024).

This work proposes a third option, challenging the prevailing assumption that accurate estimation of the curvature matrix requires private or public data at all. Leveraging the block-diagonal structure of KFAC (Dangel et al., 2025) and deep signal propagation principles (Schoenholz et al., 2016; Yang & Schoenholz, 2017), we posit that the essential optimization geometry is an intrinsic property of the model architecture, not the specific data manifold.

**DP-KFC** recovers second order statistics required to precondition gradients without accessing private training data or public datasets. The network is probed with synthetic inputs, built with random noise constructed to follow modality-specific frequency statistics (e.g., $1/f$ spectrum), that carry no privacy cost: we employ colored noise for images to simulate local and global dependencies, and we use random token sequences for text. Synthetic probing reconstructs per-layer KFAC preconditioning matrices applied to the private gradients *before* the privacy mechanism. This approach aligns with the *scale-then-privatize* design principle of Ganesh et al. (2025) while removing the dependency on private gradient statistics entirely. Our contributions are:

- **Decoupled Geometry:** The Fisher Information Matrix decouples into *Architectural Scaling* (recoverable via synthetic noise) and *Input Correlation* (recoverable via general frequency statistics of the modality), enabling preconditioner construction without spending privacy budget.

- **Privacy-Free Preconditioning:** An algorithm estimates KFAC factors using synthetic noise, requiring no private or public data, which preconditions the loss landscape to match the isotropic privacy noise.

- **Robustness to Negative Transfer:** Public data proxies fail under domain mismatch; our method strictly dominates these baselines, offering a domain-agnostic approach for private learning.

## 2. Related Work

**The Geometry of Differentially Private Optimization.** The isotropy mismatch between anisotropic loss landscapes and isotropic DP-SGD noise has motivated various mitigation strategies: adaptive clipping (Thakkar et al., 2019; Andrew et al., 2021) dynamically adjusts clipping thresholds, while Rényi DP (Mironov, 2017) provides tighter composition bounds. Yet, ill-conditioning remains a fundamental bottleneck, as the noise-to-signal ratio grows with model dimensionality, limiting the scalability of private deep learning.

**Private First- and Second-Order Methods.** In non-private settings, optimization can be accelerated by transforming the gradient to mitigate the ill-conditioned parameter space. The majority of prior attempts in DP focus on *first-order adaptive methods*, such as DP-Adam or DP-RMSProp. These algorithms attempt to capture local geometry by estimating diagonal second-moment statistics from private data. However, as shown in Ganesh et al. (2025), creating a private preconditioner involves either estimating the statistics which consumes a significant portion of the privacy budget, or using a noisy preconditioner that introduces instability often yielding performance worse than non-adaptive DP-SGD (Chen et al., 2020). A complementary line of work instead corrects the update *after* privatization: DP-AdamBC (Tang et al., 2024) removes the DP-induced bias in Adam's second-moment estimate, and DiSK (Zhang et al., 2025) denoises the privatized gradient with a simplified Kalman filter. These post-privatization corrections are orthogonal to the *scale-then-privatize* (pre-privatization) preconditioning we study; we show in Section 6.3 that the two combine favorably.

Translating *true second-order methods* (like KFAC Martens & Grosse (2015) or Newton's method Ganesh et al. (2023)) to DP introduces a more severe *dimensionality bottleneck*. Since the curvature matrix (Hessian) has dimensions $d \times d$, the number of sensitive entries scales quadratically with model size. Naive approaches that inject independent noise into the Hessian estimation suffer from a noise scale proportional to $d$ (Avella-Medina et al., 2023). Injecting sufficient noise to privatize the curvature matrix compromises its accuracy and condition number, thus the inverse preconditioner becomes unstable, amplifying noise rather than correcting the loss landscape (Ganesh et al., 2023).

**Optimization with Side Information.** To bypass these problems in privacy-preserving estimation, recent methods leverage non-sensitive side information to approximate the loss geometry. A prevalent source of side information is public data. Recent theoretical advances have established that access to unlabeled public data allows for computationally efficient private learning by reducing the problem to standard non-private optimization, even under bounded distribution shifts (Block et al., 2024). Methods like PDA-DPMD (Amid et al., 2022) and AdaDPS (Li et al., 2022) leverage this by estimating preconditioners or mirror maps on public proxies to transform private gradients prior to clipping and noise injection, while delayed-preconditioner DP-Adam (Li et al., 2023) reduces preconditioner noise by updating second-moment estimates infrequently. However, this reliance on public data is increasingly being identified as a potential concern. Tramèr et al. (2024) argue that

large-scale web data often contains sensitive information, violating the initial privacy focus, and fails to generalize to the highly specialized domains (e.g., medical imaging) where DP is most critical.

**Deep Signal Propagation and Architectural Priors.** Our approach builds on Mean Field Theory (MFT) for deep networks (Poole et al., 2016; Schoenholz et al., 2016), which shows that activation and gradient variances evolve according to deterministic maps governed by architecture (depth, initialization, non-linearity) rather than data (Yang & Schoenholz, 2017). This perspective has enabled data-independent initialization schemes ensuring trainability of deep networks without skip connections (Xiao et al., 2018). We extend MFT applications to DP for data-independent preconditioning of private gradients.

# 3. Preliminaries & Setup

We consider a deep neural network $f_\theta : \mathcal{X} \to \mathcal{Y}$ composed of $L$ layers and parameterized by $\theta = \{W_l, b_l\}_{l=1}^L$. For an input sample $x \in \mathcal{X}$, the forward propagation is defined recursively as:

$$a_0 = x, \quad s_l = W_l a_{l-1} + b_l, \quad a_l = \phi_l(s_l), \quad l = 1, \ldots, L,$$

where $W_l \in \mathbb{R}^{d_l \times d_{l-1}}$ is the weight matrix, $b_l \in \mathbb{R}^{d_l}$ the bias vector, and $\phi_l$ a component-wise nonlinearity. Given a dataset $\mathcal{D} = \{(x_i, y_i)\}_{i=1}^N$ and a loss $\ell(\cdot, \cdot)$, the empirical risk is $\mathcal{L}(\theta) = \frac{1}{N} \sum_{i=1}^N \ell(f_\theta(x_i), y_i)$. We denote the error signal (gradient with respect to pre-activations) at layer $l$ for sample $i$ as $\delta_{l,i} = \nabla_{s_l} \ell(f_\theta(x_i), y_i)$, so that the per-sample gradient with respect to $W_l$ is $\nabla_{W_l} \ell(f_\theta(x_i), y_i) = \delta_{l,i} a_{l-1,i}^\top$. When the inputs are synthetic rather than drawn from the private dataset, the resulting activations, pre-activations, and error signals are denoted with a tilde, e.g., $\tilde{a}_l$, $\tilde{s}_l$, and $\tilde{\delta}_l$.

## 3.1. Differential Privacy Framework

A randomized mechanism $\mathcal{M}$ satisfies $(\epsilon, \delta)$-Differential Privacy (Dwork, 2006) if for any two adjacent datasets $\mathcal{D}, \mathcal{D}'$ and measurable set $S$,

$$\Pr[\mathcal{M}(\mathcal{D}) \in S] \leq e^\epsilon \Pr[\mathcal{M}(\mathcal{D}') \in S] + \delta.$$

This definition imposes a strict geometric constraint: the mechanism must obscure the contribution of any individual sample vector within a sphere of uncertainty.

In deep learning, Differentially Private Stochastic Gradient Descent (DP-SGD) (Abadi et al., 2016) bounds influence via per-sample gradient clipping and injects Gaussian noise. For per-sample gradients $g^{(i)}$, clipping threshold $C$, and noise multiplier $\sigma$, the private gradient estimate is

$$\tilde{g} = \frac{1}{|\mathcal{B}|} \left( \sum_{i \in \mathcal{B}} \text{Clip}_C(g^{(i)}) + \mathcal{N}(0, \sigma^2 C^2 I) \right),$$

where $\text{Clip}_C(g) = g / \max(1, \|g\|_2 / C)$. The $L_2$ sensitivity of the sum of clipped gradients is $C$, so the Gaussian mechanism with variance $\sigma^2 C^2$ provides $(\epsilon, \delta)$-DP guarantees under standard composition theorems. Privacy accounting is performed on the noisy sum; the subsequent averaging scales both signal and noise uniformly.

## 3.2. Gradient Preconditioning and KFAC

To address the anisotropic geometry of the loss landscape, Natural Gradient Descent (NGD) (Amari, 1998) updates parameters using $\theta_{t+1} = \theta_t - \eta F^{-1} \nabla_\theta \mathcal{L}$, where $F$ is the Fisher Information Matrix (FIM). However, for differentially private optimization, we instead use *gradient preconditioning* with $F^{-1/2}$:

$$\theta_{t+1} = \theta_t - \eta F^{-1/2} \nabla_\theta \mathcal{L}(\theta_t).$$

This transformation makes the preconditioned gradient $\tilde{g} = F^{-1/2} g$ approximately isotropic ($\mathbb{E}[\tilde{g}\tilde{g}^\top] \approx I$), which is essential for DP: the spherical clipping and isotropic noise addition of DP-SGD become geometrically matched in this transformed space (Appendix B).

To make preconditioning tractable, we use the KFAC approximation (Martens & Grosse, 2015), assuming independence between activations and errors. Each layer-wise FIM block is approximated as a Kronecker product

$$F_l \approx A_{l-1} \otimes G_l, \quad A_{l-1} = \mathbb{E}[a_{l-1} a_{l-1}^\top], \quad G_l = \mathbb{E}[\delta_l \delta_l^\top],$$

allowing efficient computation of $F_l^{-1/2} = A_{l-1}^{-1/2} \otimes G_l^{-1/2}$.

## 3.3. Mean-Field Signal Propagation

Following Poole et al. (2016); Schoenholz et al. (2016), in the infinite-width limit the pre-activations $s_l$ can be modeled as Gaussian with variance $q^l = \mathbb{E}[s_l^2]$. This variance evolves according to the deterministic map

$$q^l = \mathcal{V}(q^{l-1} \mid \sigma_w, \sigma_b) = \sigma_w^2 \int \phi(\sqrt{q^{l-1}} z)^2 \mathcal{D}z + \sigma_b^2,$$

where $\mathcal{D}z$ is the standard Gaussian measure. For standard initializations, $q^l$ converges exponentially to a stable fixed point $q^*$, making the deep layer variance independent of the input variance $q^0$ (Poole et al., 2016).

Similarly, the backward error variance $\tilde{q}^l = \mathbb{E}[\|\delta_l\|^2]$ evolves according to a dual recursion governed by $\phi'$, determining whether gradients vanish or explode (Schoenholz et al., 2016). As shown by Karakida et al. (2019), the trace of the layer-wise Fisher block is proportional to the product of these variances:

$$\text{Tr}(F_l) \propto d \cdot q^{l-1} \cdot \tilde{q}^l, \tag{1}$$

indicating that the overall magnitude of the preconditioner matrix (its trace) is determined by architecture ($\sigma_w, \sigma_b, \phi$)

rather than data. While this result is derived in the infinite-width limit, we empirically validate in Section 6.2 that the spectral alignment holds for practical finite-width architectures.

# 4. DP-KFC

We now introduce DP-KFC, a second-order private optimization method whose core components are: *(i)* a data-free KFC preconditioner, *(ii)* synthetic inputs that recover the relevant architectural and data statistics, and *(iii)* integration of the preconditioner into a differentially private update following the scale-then-privatize principle.

## 4.1. Synthetic Preconditioning

Since the overall magnitude of the preconditioner matrix is an intrinsic property of the architecture, the parameter sensitivities can be recovered without accessing private data. We therefore construct the KFC preconditioning blocks probing the network with structurally coherent data but semantically independent to the private dataset (Algorithm 1).

We first generate a synthetic batch $\mathcal{B}_{\text{syn}} = \{\tilde{x}^{(j)}, \tilde{y}^{(j)}\}_{j=1}^M$ using modality-specific noise distributions (see Section 4.2 and Appendix E). A forward and backward pass is then performed with the current network parameters $\theta_t$ to obtain feature activations $\tilde{a}_{l-1}$ and error signals $\tilde{\delta}_l$ for each layer. Using the outer-product structure of per-sample gradients, we aggregate these quantities into layer-wise Kronecker covariance factors, $\hat{A}_{l-1} = \mathbb{E}[\tilde{a}_{l-1}\tilde{a}_{l-1}^\top]$ and $\hat{G}_l = \mathbb{E}[\tilde{\delta}_l\tilde{\delta}_l^\top]$, with an added damping term $\pi I$ to floor the low-variance directions (Martens & Grosse, 2015).

To normalize the anisotropic parameter sensitivity induced by the network, we compute the regularized inverse square roots of $\hat{A}_{l-1}$ and $\hat{G}_l$ via eigendecomposition, yielding matrices $U_{A,l}$ and $U_{G,l}$, with a term $\gamma I$ for numerical stability. The resulting preconditioner $\tilde{F}^{-1/2}$ is implicitly given by the Kronecker product $U_A \otimes U_G$. This formulation avoids materializing the full FIM, whose dimensionality would be prohibitive.

## 4.2. Synthetic Data Generation

Creating the synthetic probes involves generating an input batch that is capable of simulating the forward (layer activations) and backward (error gradients) pass of private data while remaining independent from it to ensure DP guarantees.

Let $A_{l-1} = Q_{A,l-1}\Lambda_{A,l-1}Q_{A,l-1}^\top$ denote the eigendecomposition of the activation covariance at layer $l$. The eigenvalues $\Lambda_{A,l-1}$ quantify the scale of each principal direction, while the eigenvectors $Q_{A,l-1}$ define correlation structures across features. Following Eq. 1, signal propagation mag-

---

**Algorithm 1** DP-KFC Preconditioner Construction (Synthetic Probing)

---

1: **Input:** Neural Network $f_\theta$, batch size $M$, damping $\pi$, stability $\gamma$
2: **Output:** Data-free preconditioner $\tilde{F} = \{\tilde{F}_l^{-1/2}\}_{l=1}^L$
3: // Synthetic input generation (independent of private data)
4: $\{\tilde{x}^{(j)}, \tilde{y}^{(j)}\}_{j=1}^M \leftarrow$ Generate_Samples$(M)$ (App. E)
5: // Forward and backward passes (see Preliminaries)
6: Compute $\{\tilde{a}_{l-1}^{(j)}, \tilde{\delta}_l^{(j)}\}$ via forward and backward propagation
7: // Layer-wise KFAC factor estimation and normalization
8: **for** each layer $l \in [L]$ **do**
9:    // Compute activation and gradients covariances
10:    $\hat{A}_{l-1} = \frac{1}{M}\sum_{j=1}^M \tilde{a}_{l-1}^{(j)}(\tilde{a}_{l-1}^{(j)})^\top + \pi I$
11:    $\hat{G}_l = \frac{1}{M}\sum_{j=1}^M \tilde{\delta}_l^{(j)}(\tilde{\delta}_l^{(j)})^\top + \pi I$
12:    // Eigen decomposition
13:    $[Q_A, \Lambda_A] \leftarrow$ EigDecomp$(\hat{A}_{l-1})$
14:    $[Q_G, \Lambda_G] \leftarrow$ EigDecomp$(\hat{G}_l)$
15:    // Curvature normalization
16:    $U_{A,l} = Q_A(\Lambda_A + \gamma I)^{-1/2}Q_A^\top$
17:    $U_{G,l} = Q_G(\Lambda_G + \gamma I)^{-1/2}Q_G^\top$
18:    $\tilde{F}_l^{-1/2} \leftarrow U_{A,l} \otimes U_{G,l}$
19: **end for**
20: **return** $\tilde{F}^{-1/2} = \{\tilde{F}_l^{-1/2}\}_{l=1}^L$

---

nitudes ($\text{Tr}(A_{l-1})$ and $\text{Tr}(G_{l-1})$) are fixed by weight variances rather than input statistics; hence, unstructured noise with random labels suffices to recover the scale of the KFAC factors. Synthetic labels $\tilde{y}$ are sampled uniformly from the label space; since KFAC factors depend on error signal magnitudes rather than label correctness, random labels suffice for estimating $G_l$.

However, the eigenvectors $Q_{A,l-1}$ define the spatial or sequential relationships among features, which are entirely absent in uncorrelated (full-rank) white noise. Moreover, white noise distributes energy uniformly across all frequencies, whereas deep networks propagate low-frequency features while suppressing high-frequency ones (Rahaman et al., 2019). To simulate these structural correlations, we exploit the statistical regularity of natural images, which universally exhibit $1/f^\alpha$ power spectra, where amplitude decays with spatial frequency (Field, 1987). This statistical regularity means input correlations can be approximated without access to private data. We explicitly construct synthetic batches following this power-law distribution ($S(f) \propto 1/f^\alpha$). For image domains, we generate Pink Noise by modulating white noise $Z \in \mathbb{C}^{H \times W}$ in the frequency domain. Let $\mathbf{u} \in \mathbb{R}^2$ be the spatial frequency coordinate vector relative to the zero-frequency component

(the center of the spectrum). We scale the spectrum as:

$$\tilde{Z}_{\mathbf{u}} = Z_{\mathbf{u}} \cdot \frac{1}{\|\mathbf{u}\|_2^{\alpha/2} + \epsilon}, \quad x_{\text{syn}} = \text{Re}(\mathcal{F}^{-1}(\tilde{Z})) \quad (2)$$

where $\|\mathbf{u}\|_2$ is the Euclidean distance from the spectral center, and $\epsilon$ prevents division by zero at the origin. Here, Pink Noise ($\alpha \approx 1$) concentrates probing energy into the low-frequency modes to match the ideal private preconditioner.

For discrete domains (NLP), inputs are token indices rather than continuous signals, so the analogue of spectral shaping is a *structural token sampling* strategy: each probe is a sequence drawn from a frequency-weighted distribution over the vocabulary (mimicking the heavy-tailed Zipfian statistics of natural text), with the model's structural tokens ([CLS], [SEP], padding) placed at their canonical positions, so that the probe exercises the same attention and LayerNorm pathways as real text without carrying its semantics. Full details of both generators are in Appendix E. As discussed in Section 6.3, this recovers the architectural factors but not the low-dimensional manifold occupied by real text, which is the main reason synthetic probing trails public-data preconditioning on NLP fine-tuning.

### 4.3. Privatizing preconditioned gradients

Once the preconditioning matrices $U_{A,l}$ and $U_{G,l}$ are computed via the data-free procedure (Algorithm 1), they are frozen and broadcast to the private training loop. For the subsequent $T_{freq}$ steps, we replace the standard gradient computation with a preconditioned gradient oracle following Algorithm 2. This operation transforms the per-sample gradients from the original parameter space, where parameter sensitivity varies by orders of magnitude, into a preconditioned coordinate system with homogeneous parameter sensitivity. Contrary to standard DP-SGD, where high-sensitivity directions are severely clipped and low-sensitivity directions are submerged by noise, in the preconditioned space, every parameter contributes equally to the norm ($Cov(\tilde{g}) = I$). Consequently, the fixed clipping threshold is applied uniformly across all parameters, preserving the update direction while upscaling low-sensitivity directions.

The transformation $\tilde{g}_l = U_{G,l} \cdot g_l \cdot U_{A,l}$ in Algorithm 2 assumes the gradient is a matrix. For convolutional layers, we apply the standard im2col transformation, treating each spatial location as an independent sample (Martens & Grosse, 2015). For Transformers, all attention and MLP components are Linear layers, so standard KFAC applies directly. Following the K-FAC-reduce formulation of Eschenhagen et al. (2023), we average covariances over the weight-sharing dimension (spatial locations for CNNs, sequence positions for attention), which is computationally efficient and empirically equivalent to the more expensive K-FAC-expand variant. Implementation details are provided

---

**Algorithm 2** DP-KFC Training Loop

1: **Input:** Private Data $\mathcal{D}$, Learning rate $\eta$, Noise scale $\sigma$, Clip $C$, Freq $T_{freq}$
2: **Output:** Private Model Parameters $\theta_T$
3: **for** $t = 0, \ldots, T - 1$ **do**
4:    // Periodic Preconditioner Update
5:    **if** $t \mod T_{freq} == 0$ **then**
6:       $\{U_{A,l}, U_{G,l}\} \leftarrow$ **Alg. 1**$(\theta_t, M, \pi, \gamma)$
7:    **end if**
8:    // Private Gradient Step
9:    Sample private batch $\mathcal{B}_{\text{priv}} \subset \mathcal{D}$
10:    **for** sample $i \in \mathcal{B}_{\text{priv}}$ **do**
11:      **for** layer $l \in L$ **do**
12:        Compute Gradient: $g_l^{(i)} = \nabla_{W_l} \ell(x_i)$
13:        **Transform:** $\tilde{g}_l^{(i)} \leftarrow U_{G,l} \cdot g_l^{(i)} \cdot U_{A,l}$
14:      **end for**
15:      Compute Norm: $\nu_i = \sqrt{\sum_l \|\tilde{g}_l^{(i)}\|_F^2}$
16:      **Clip:** $\bar{g}^{(i)} \leftarrow \tilde{g}^{(i)} / \max(1, \nu_i/C)$
17:    **end for**
18:    // 3. Noise Addition and Descent
19:    $\tilde{G} \leftarrow \frac{1}{|\mathcal{B}|} \left( \sum_i \bar{g}^{(i)} + \mathcal{N}(0, \sigma^2 C^2 I) \right)$    // Averaging
20:    $\theta_{t+1} \leftarrow \theta_t - \eta \cdot \tilde{G}$
21: **end for**

---

in Appendix E.3.

## 5. Convergence Analysis

This section provides a convergence analysis for DP-KFC on non-convex objectives, the standard setting for deep neural networks. In DP-KFC, private gradients are normalized, based on the inherent parameter sensitivity estimates, before injecting noise. Thus, reducing the condition number governing the convergence rate. We now analyze the proposed method following standard assumptions for stochastic non-convex optimization.

**Assumption 5.1** (*L*-Smoothness). The loss function $\mathcal{L}(\theta)$ is differentiable and $L$-smooth, i.e., $\|\nabla\mathcal{L}(\theta_1) - \nabla\mathcal{L}(\theta_2)\| \leq L\|\theta_1 - \theta_2\|$ for all $\theta_1, \theta_2 \in \mathbb{R}^d$.

**Assumption 5.2** (Bounded Variance). The stochastic gradient oracle is unbiased, $\mathbb{E}[\nabla\ell(x_i)] = \nabla\mathcal{L}(\theta)$, with bounded variance $\sigma_{sgd}^2$. Since preconditioning is applied before clipping (Algorithm 2), gradient norms are homogenized prior to the clipping operation. This reduces the fraction of gradients exceeding the threshold $C$, minimizing clipping-induced bias compared to standard DP-SGD.

**Assumption 5.3** (Bounded Preconditioner). The noise-derived preconditioner $P_t$ (constructed via preconditioning matrices $U_G, U_A$) has bounded eigenvalues: $0 < \lambda_{min} I \preceq P_t \preceq \lambda_{max} I$. This is structurally guaranteed in our algorithm by the Tikhonov damping (regularization) terms $\pi I$ and $\gamma I$ added to the covariance matrices before inversion.

To establish the convergence guarantee, these structural properties are essential. Smoothness (Assumption 5.1) bounds the loss reduction via a quadratic approximation. Assumption 5.2 separates the estimation variance from the privacy noise. Finally, the spectral bounds in Assumption 5.3 are critical: the lower bound guarantees that the preconditioned update remains a valid descent direction, while the upper bound limits the amplification of stochastic variance.

## 5.1. Convergence Rate

The following theorem characterizes the convergence rate to a critical point. We separate the geometric scaling of the variance coming from the gradient stochastic noise and the additive nature of the privacy noise.

**Theorem 5.4** (Non-Convex Convergence of DP-KFC). *Let the learning rate be $\eta = \frac{1}{\sqrt{T}}$. After $T$ iterations with batch size $B$, the algorithm yields a solution sequence $\{\theta_t\}$ satisfying:*

$$\min_{t \in [0,T-1]} \mathbb{E}[\|\nabla \mathcal{L}(\theta_t)\|^2] \leq \frac{C_1}{\lambda_{min}\sqrt{T}} +$$
$$\frac{C_2}{\lambda_{min}\sqrt{T}} \left( \lambda_{max}^2 \sigma_{sgd}^2 + \frac{d\sigma^2 C^2}{B^2} \right) \quad (3)$$

*where $C_1$ depends on the initial loss gap $\mathcal{L}_0 - \mathcal{L}^*$, and $C_2$ depends on the smoothness constant $L$.*

Eq. (3) shows the advantage over standard preconditioned DP-SGD. In methods that directly precondition noisy gradients ($P_t(\bar{g}_t + \xi_t)$), the privacy noise is multiplied by the preconditioner, amplifying the privacy error variance by $\lambda_{max}^2$. In contrast, our approach decouples these stages: because $P_t$ is batch-independent, we reshape the gradient before the privacy mechanism applies. Consequently, the privacy term $d\sigma^2 C^2/B^2$ avoids being amplified by $\lambda_{max}^2$. The eigenvalue terms $\lambda_{min}^{-1}$ and $\lambda_{max}^2$ in the bound are controlled by the damping parameter $\gamma$ in Algorithm 1: eigenvalues satisfy $\lambda_{min} \geq \sqrt{\gamma}$ and $\lambda_{max} \leq 1/\sqrt{\gamma}$ for normalized co-variances. With typical $\gamma = 10^{-2}$, the effective condition number remains moderate. In high-privacy regimes where privacy noise dominates, avoiding its amplification by $\lambda_{max}^2$ yields the empirical gains. The $\mathcal{O}(T^{-1/2})$ rate is the optimal rate class for first-order stochastic non-convex optimization (Arjevani et al., 2023); as is standard for second-order methods in the non-convex regime, preconditioning improves the *constants* (a smaller effective condition number) rather than the rate, so the bound is informative precisely when the realized preconditioner is well conditioned. A full derivation is provided in Appendix A, and complexity analysis in Appendix D.

**Why the realized preconditioner is well conditioned.** The bound depends on the spectrum of the *realized* map $\tilde{F}^{-1/2}F\tilde{F}^{-1/2}$, which is small precisely because the syn-

thetic and private curvatures share their eigenspectrum. Building on the mean-field variance contraction of Section 3.3 (formalized in Proposition F.1), Theorem F.2 (Appendix F) shows that the eigenspectra of the private and synthetic KFAC factors are proportional in deep networks, so preconditioning with $\tilde{F}^{-1/2}$ collapses the effective curvature toward a near-identity (scalar) operator, with the residual scalar absorbed by the clipping norm and learning rate. Section 6.2 and Fig. 2 confirm this alignment empirically across MLP, CNN, and attention layers; Appendix I further characterizes how performance degrades gracefully as the synthetic prior is mismatched.

*Remark* 5.5 (The privacy wall). Unlike non-private training, where compute bounds the number of useful steps, DP training has an intrinsic horizon: privacy accounting fixes the number of steps $T$ achievable at a target $(\epsilon, \delta)$, since each step spends budget. Per-step wall-clock overhead (here $\approx 2.2\times$ lower sample throughput than DP-SGD; Appendix D) is therefore traded against *convergence within a fixed step count*: a method that makes each of the $T$ permitted steps more effective is preferable even if individual steps are slower, because extra steps are not an option. This reframes the cost of second-order preconditioning under DP and motivates spending compute on better geometry rather than on more iterations.

## 5.2. Privacy Analysis

The privacy guarantee of DP-KFC relies on the preconditioner $P_t$ being batch-independent: it is computed from synthetic noise and the current model state $\theta_t$, which depends only on *past* privatized releases, not the current batch $\mathcal{B}$. This ensures that the $L_2$ sensitivity of the preconditioned, clipped gradient sum remains bounded by $C$.

### 5.2.1. FORMAL GUARANTEE

The following proposition establishes that applying a batch-independent linear transformation prior to the Gaussian mechanism preserves the standard differential privacy bounds.

**Proposition 5.6** (Privacy of Noise-Preconditioned Updates). *Let $\mathcal{M}_{Gauss}(\mathbf{x}) = \sum \mathbf{x}_i + \mathcal{N}(0, \sigma^2 C^2 I)$ be the standard Gaussian mechanism with $L_2$ sensitivity $C$. Let $P_t$ be a linear operator that is independent of the current batch $\mathcal{B}$ (computed from synthetic noise and model parameters $\theta_t$ derived from past privatized updates). The composite mechanism:*

$$\mathcal{M}_{KFC}(\mathcal{B}) = \sum_{i \in \mathcal{B}} Clip_C(P_t \cdot \nabla \ell(x_i)) + \mathcal{N}(0, \sigma^2 C^2 I) \quad (4)$$

*satisfies the same $(\alpha, \epsilon)$-Rényi Differential Privacy (RDP) (Mironov, 2017) guarantees as $\mathcal{M}_{Gauss}$ (Appendix C).*

DP-KFC can be reduced to a form of implicit adaptive clip-

ping. By transforming the gradient with $P_t$ prior to the fixed clipping operation, we dynamically align the clipping boundary with the local geometry of the loss landscape. This allows us to normalize gradient scales, ensuring significant updates are not disproportionately clipped, without the need to estimate gradient norms from private data. Consequently, we obtain the benefits of adaptive clipping (Andrew et al., 2021) while retaining the simple, tight accounting of standard fixed-threshold mechanisms, as the adaptation is derived entirely from non-sensitive architectural properties.

Having established the theoretical foundations for convergence and privacy preservation, we now validate these properties empirically across diverse tasks and privacy regimes.

# 6. Experiments

This section evaluates DP-KFC with Synthetic Preconditioning across computer vision and natural language processing tasks under varying privacy regimes. Three sets of experiments are shown. First, *Validation of Architectural Dominance* (Section 6.2): parameter sensitivity and the eigenspectrum of the KFAC factors can be effectively extracted using only model architecture and synthetic noise. Second, *Benchmark Performance across Modalities* (Section 6.3): data-free preconditioning consistently outperforms standard DP-SGD. Third, *Transfer and Domain Mismatch* (Section 6.4): which scenarios benefit from public data versus synthetic noise, depending on domain alignment. Ablation studies confirm robustness to hyperparameter variations and structured noise compositions (Appendix I).[1]

## 6.1. Experimental Setup

**Implementation and Hardware.** All experiments were implemented in PyTorch using the Opacus library (Yousefpour et al., 2021) for differential privacy accounting. Training was conducted on NVIDIA A100 GPUs. All results are averaged over 10 independent seeds; shaded regions in figures and $\pm$ values in tables denote one standard deviation.

**Models and Tasks.** We evaluate across four tasks: CNN on MNIST, CrossViT on CIFAR-100, BERT on StackOverflow, and logistic regression on IMDB. For Transformer architectures, we use AdamW as it outperforms SGD, making the DP-SGD baseline equivalent to DP-Adam. Full benchmark details in Appendix H. Importantly, hyperparameter analysis (Appendix I) shows that KFAC-specific parameters $(\gamma, \alpha)$ have low importance and can use defaults without significant performance degradation.

[1] Code is publicly available at https://github.com/molinamarcvdb/DP-KFC.

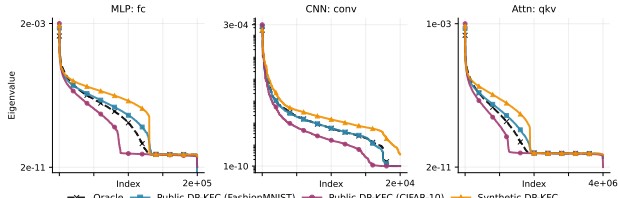

*Figure 2.* **Eigenspectrum Alignment.** Sorted eigenvalues of KFAC factors for distinct layers: (a) MLP fully-connected, (b) CNN convolutional, and (c) Attention QKV projection. Domain-matched public data (FashionMNIST, blue) aligns closely with the private oracle (black, dashed). Domain-mismatched data (CIFAR-10, purple) shows larger deviation. Synthetic DP-KFC (orange) captures the eigenvalue decay across architecture types.

## 6.2. Validation of Architectural Dominance

Architectural dominance is validated by comparing eigenspectra of the KFAC factors derived from three sources: private data (oracle), public data (proxies), and synthetic noise (ours). For each source, we estimate the covariance factors $\hat{A}_l$ and $\hat{G}_l$, then compute their eigendecompositions. Since the layer-wise preconditioner is the Kronecker product $F_l^{-1} \approx A_l^{-1} \otimes G_l^{-1}$, the full spectrum is reconstructed as $\lambda_F = \lambda_A \otimes \lambda_G$. To track alignment throughout training, we measure *cosine similarity* $\text{CosSim}(C^*, \hat{C}) = \text{Tr}(C^{*\top}\hat{C})/(\|C^*\|_F\|\hat{C}\|_F)$ for directional alignment and *relative Frobenius error* $\text{RelErr} = \|C^* - \hat{C}\|_F/\|C^*\|_F$ for magnitude error (Kunstner et al., 2019).

Fig. 2 confirms that Synthetic DP-KFC recovers the eigenvalue decay of representative MLP, CNN and Attention layers, matching public data baselines. The alignment extends to attention-based architectures, where the QKV projection layer, despite having millions of eigenvalues, exhibits the same architecture-determined spectral decay. In Appendix F we provide the complete layer-wise spectral analysis and validate these findings further using Stochastic Lanczos Quadrature (Ghorbani et al., 2019) to estimate the full Hessian density.

Beyond static alignment, the synthetic estimator tracks the private eigenspectrum throughout training, not merely at initialization. Fig. 3 shows that in early layers, Synthetic DP-KFC maintains cosine similarity above 0.8 with the oracle, comparable to matched public data, while mismatched data stagnates around 0.6. Scale error of Synthetic DP-KFC remains lower than public proxies throughout. In deeper layers, directional alignment degrades across all methods ($< 0.6$), indicating eigenvector directions depend on private labels. However, Synthetic DP-KFC achieves superior scale stability with lower error magnitude than public proxies. Full per-layer metrics in Appendix G. The same directional ranking extends to attention-based architectures (Appendix G.3), though the scale advantage of synthetic noise is reduced by LayerNorm, which bounds activation magnitudes across sources.

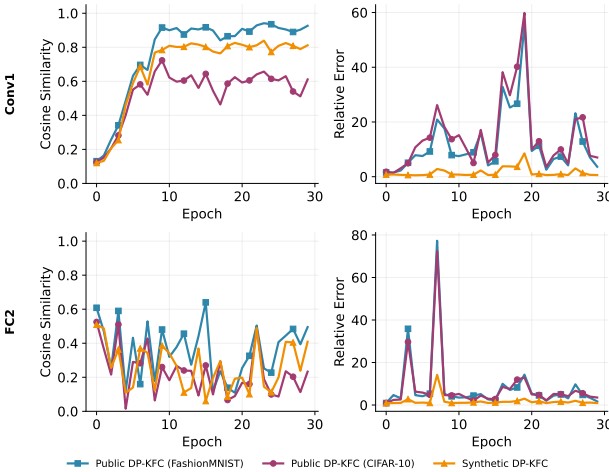

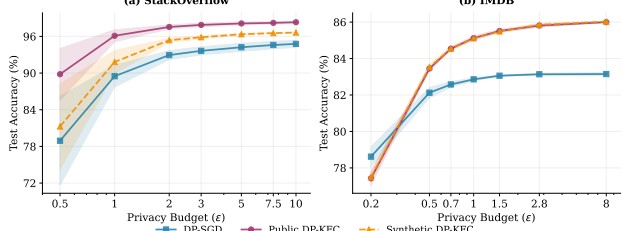

*Figure 5.* **NLP Tasks.** Test accuracy vs. $\epsilon$ on (a) StackOverflow next-word prediction (BERT) and (b) IMDB sentiment classification (logistic regression); baseline is DP-Adam / DP-SGD respectively, dashed lines the non-private references ($\approx 99.0\%$, $\approx 88.0\%$). Synthetic DP-KFC (orange) matches public-data preconditioning on IMDB; on StackOverflow it improves over the baseline but trails Public DP-KFC (analyzed below). Same $\epsilon$-axis convention as Fig. 4.

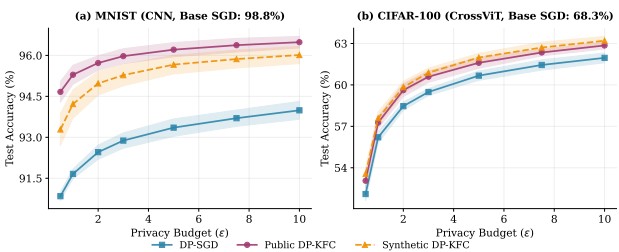

*Figure 3.* **Similarity of Public and Synthetic Preconditioning with Oracle.** (a) Cosine similarity (Direction) and (b) Relative Frobenius norm (Scale) comparing FashionMNIST (Blue), CIFAR-10 (Purple), and Synthetic DP-KFC (Orange) against the private oracle. Top row: First convolutional layer. Bottom row: Deepest layer where directional alignment is low and not properly recovered by any preconditioning method.

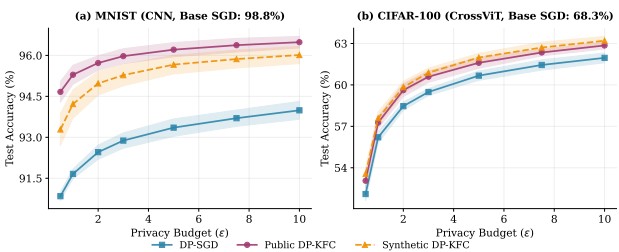

*Figure 4.* **Privacy-Utility Trade-off.** Test accuracy vs. privacy budget $\epsilon$ for DP-SGD (Blue; DP-Adam for CrossViT), Public DP-KFC (Purple), and Synthetic DP-KFC (Red). Synthetic DP-KFC consistently matches or exceeds public-data baselines without distribution-shift cost. See text for the $\epsilon$-axis convention.

### 6.3. Benchmark Performance across Modalities

We compare Synthetic DP-KFC against two baselines: (i) standard DP-SGD, representing first-order private optimization, and (ii) Public DP-KFC, which constructs preconditioners from public proxy data. Results span four tasks across vision (MNIST, CIFAR-100) and language (StackOverflow, IMDB) modalities, with privacy budgets $\epsilon \in [0.5, 10.0]$ (Figs. 4–5). In these figures each marker is a *separate* model whose target $\epsilon$ is the end-of-training budget: the noise multiplier $\sigma$ is calibrated by RDP accounting over the $T$ steps, and all hyperparameters are tuned per method and per (dataset, $\epsilon$) pair; non-DP references appear as dashed lines. Full experimental details and per-$\epsilon$ results are in Appendix H.

**Computer Vision.** Both Public and Synthetic DP-KFC consistently outperform DP-SGD across privacy regimes. On MNIST at $\epsilon = 1.0$, Synthetic DP-KFC achieves 94.2% versus 91.7% for DP-SGD (+2.5%). On CIFAR-100 with

CrossViT, Synthetic DP-KFC matches Public DP-KFC across all $\epsilon$ values, demonstrating that architecture alone can be used to estimate the preconditioning matrix even for attention-based models without public data.

**Natural Language.** Fig. 5 confirms the trend extends to NLP. On StackOverflow with BERT, Synthetic DP-KFC reaches 91.8% at $\epsilon = 1.0$ compared to 89.5% for DP-SGD, despite using semantically meaningless token probes. On IMDB with logistic regression, both Public and Synthetic DP-KFC achieve 85.8% versus 83.1% for DP-SGD at $\epsilon = 2.8$.

**When the data manifold matters.** Gains are largest when training from scratch (MNIST +2.5%, IMDB +2.7%), where preconditioning keeps low-sensitivity layers from drowning in noise; in frozen-backbone fine-tuning (CIFAR-100 $\approx +1.4\%$) the pre-trained features are already well conditioned (Martens & Grosse, 2015; Karakida et al., 2019), leaving less to correct. The synthetic prior also matches public data exactly on vision, since natural images obey $1/f^{\alpha}$ spectra (Field, 1987) and pink noise lands near their activation statistics, but not on language: real text occupies a low-dimensional manifold in the embedding space, so random token probes recover the architectural factors yet land off it, leaving a gap on StackOverflow, where Public DP-KFC reaches 96.1% at $\epsilon = 1$ (Appendix H). Closing this, e.g. with token-frequency priors or embedding-space probes, is left to future work.

**Standalone and complementary.** DP-KFC follows the *scale-then-privatize* (STP) principle (Ganesh et al., 2025), extending it from diagonal (Li et al., 2022; 2023) to block-diagonal KFAC geometry at zero privacy cost. Among data-free methods on MNIST it beats both that STP baseline (AdaDPS) and post-privatization methods acting after the noise, namely DP-AdamBC (Tang et al., 2024) (bias correction) and DiSK (Zhang et al., 2025) (Kalman denoising); see Table 1. DP-KFC with public data improves further. The two families are complementary: *Synthetic DP-KFC*

*Table 1.* **Comparison with private optimizers** on MNIST (CNN, test accuracy %, 5 seeds, hyperparameters tuned per method). Top block: standalone optimizers (Public DP-KFC additionally requires a public proxy). Bottom: pre-privatization STP combined with post-privatization bias correction. Bold marks the best entry per column. Full $\epsilon$ range in Appendix H.2.

| Method | $\epsilon=1$ | $\epsilon=2$ | $\epsilon=8$ |
|---|---|---|---|
| DP-SGD | 91.7±0.2 | 92.5±0.3 | 93.7±0.3 |
| AdaDPS (Li et al., 2022) | 91.3±0.8 | 93.2±1.0 | 93.3±1.4 |
| DiSK (Zhang et al., 2025) | 93.7±0.4 | 94.1±0.3 | 94.3±0.2 |
| DP-AdamBC (Tang et al., 2024) | 94.0±0.3 | 94.8±0.2 | 95.3±0.1 |
| Public DP-KFC | 95.3±0.4 | 95.7±0.3 | 96.4±0.3 |
| **Synthetic DP-KFC** | 94.2±0.5 | 95.0±0.4 | 95.9±0.3 |
| **Synthetic DP-KFC + DP-AdamBC** | **95.5**±0.3 | **96.1**±0.2 | **96.4**±0.3 |

*+ DP-AdamBC* beats either alone (e.g. 95.5% vs. 94.2% and 94.0% at $\epsilon=1$), a direction we leave open. Details in Appendix H.2.

### 6.4. Transfer and Domain Mismatch

To quantify robustness against distribution shift, we evaluate across two transfer regimes representing different levels of domain mismatch between the proxy (source) and the private task (target). We compare against AdaDPS (Li et al., 2022), which constructs adaptive preconditioners for DP optimization. We include two variants: *(i)* AdaDPS (Public) using MNIST as proxy, and *(ii)* AdaDPS (Pink) replacing public data with synthetic pink noise to isolate the preconditioning mechanism from the data source.

*(i) Ideal Alignment* (Fashion ← MNIST): Source and target share high structural similarity, centered objects, black backgrounds, and equal resolution. Both datasets contain single-channel grayscale images with similar spatial statistics, making MNIST an informative geometric proxy for FashionMNIST.

*(ii) Texture Disjoint* (Path ← MNIST): A challenging scenario where object-centric priors (handwritten digits) conflict with texture-centric medical histology tasks. PathMNIST contains colorectal cancer tissue patches with fundamentally different visual statistics, evaluating susceptibility to *Negative Transfer*.

Table 2 summarizes the results across these scenarios. The *Oracle (Private)* baseline computes the KFAC preconditioner directly from private data, representing the upper bound achievable with unlimited privacy budget for curvature estimation.

**Negative Transfer.** Synthetic DP-KFC emerges as a robust geometric prior across both transfer regimes. Under *Ideal Alignment*, it nearly matches the Private Oracle (87.8% vs 88.3%), outperforming both DP-SGD (83.5%) and AdaDPS variants (84.7%, 84.2%). More importantly, under *Texture*

*Table 2.* **Transfer Robustness.** Test accuracy (%) under domain mismatch at $\epsilon = 1.0$. Oracle uses private data.

| Method | Ideal Align. (Fashion←MNIST) | Texture Disj. (Path←MNIST) |
|---|---|---|
| Oracle (Private) | 88.3±0.2 | 78.4±1.7 |
| DP-SGD | 83.5±0.7 | 68.5±2.3 |
| AdaDPS (Public) | 84.7±0.3 | 70.5±2.0 |
| AdaDPS (Pink) | 84.2±0.5 | 71.2±1.9 |
| Public DP-KFC | 87.6±0.2 | 73.4±1.3 |
| **Synthetic DP-KFC** | **87.8**±0.2 | **78.2**±1.9 |

*Disjoint*, where the MNIST proxy's object-centric statistics conflict with PathMNIST's texture patterns, Synthetic DP-KFC achieves 78.2%, matching the Oracle while the Public DP-KFC degrades to 73.4%. This 4.8% gap demonstrates that synthetic noise captures task-agnostic geometric structure (e.g., spatial correlations, layer-wise scaling) without inheriting domain-specific biases that cause negative transfer. The consistent performance suggests synthetic probing acts as implicit regularization, capturing global architectural properties while avoiding high-variance batch-specific estimates (Appendix G).

## 7. Conclusion

This work challenges the assumption that estimating the Fisher Information Matrix requires semantic data: gradient *scale* is an intrinsic architectural property while eigenvector *directions* are data-dependent, so the curvature needed for preconditioning is recoverable from synthetic noise, with no public data and no privacy budget. This makes private model development feasible in specialized domains (e.g., medical imaging) that lack public proxies, and lets federated clients build local preconditioners without communicating second-order statistics.

DP-KFC incurs the usual KFAC overhead ($\approx 2.2\times$ lower sample throughput; Appendix D), typically offset by faster convergence within DP's fixed step budget (Remark 5.5). Directional alignment degrades in deep layers, and on NLP synthetic token probes miss the low-dimensional text manifold, leaving a gap to public-data preconditioning (Section 6.3). Future work includes low-rank approximations, manifold-aware synthetic generators for language, combining pre- and post-privatization corrections (e.g., with DP-AdamBC (Tang et al., 2024)), and extending the *scale-then-privatize* paradigm to other preconditioned optimizers such as Muon and Shampoo (Jordan et al., 2024; Gupta et al., 2018).

## Impact Statement

This work aims to advance privacy-preserving machine learning by enabling second order optimization methods under differential privacy without requiring auxiliary public data. Removing the dependence on public proxies, our method expands the applicability of private deep learning to sensitive domains, such as medical imaging and financial data, where suitable public data is unavailable or itself contains private information, or labels are time consuming. We do not foresee specific negative societal consequences beyond those inherent to advances in machine learning broadly. The privacy guarantees provided by differential privacy remain formally intact under our method, thus pushing the privacy-utility frontier.

## Acknowledgements

This project is supported by the CAFEIN® research and development fund. This project is also funded by the Innovative Health Initiative Joint Undertaking and its members under grant agreement No. 101172825. Funded by the European Union, the private members, and those contributing partners of the IHI JU. Views and opinions expressed are however those of the author(s) only and do not necessarily reflect those of the aforementioned parties. Neither of the aforementioned parties can be held responsible for them. We further acknowledge the support of the CERN Quantum Technology Initiative (QTI). This work was supported by the María de Maeztu Units of Excellence Programme (CEX2021-001195-M), funded by the Spanish Government (MICIU/AEI/10.13039/501100011033), and by the European Union under the ERC Synergy Grant *Zee-Zoom-Zap* (grant no. 101224844). Views and opinions expressed are however those of the author(s) only and do not necessarily reflect those of the European Union or the European Research Council Executive Agency; neither the European Union nor the granting authority can be held responsible for them.

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

# A. Convergence Analysis

In this section, we provide the complete proof for the convergence of DP-KFC to a stationary point for non-convex objectives. We begin by defining the notation and probabilistic setup used throughout the analysis.

## A.1. Notation and Problem Setup

We consider the optimization of a differentiable, non-convex loss function $\mathcal{L}(\theta) : \mathbb{R}^d \to \mathbb{R}$. We denote the optimal loss as $\mathcal{L}^* = \inf_\theta \mathcal{L}(\theta) > -\infty$.

At each iteration $t$, the algorithm performs the following update:

$$\theta_{t+1} = \theta_t - \eta_t \left( P_t \cdot \bar{g}_t + \xi_t \right) \tag{5}$$

Table 3 summarizes the random variables and constants involved in this process.

*Table 3.* Summary of Notation for Convergence Analysis

| Symbol | Description | Properties / Assumptions |
|--------|-------------|--------------------------|
| $\theta_t$ | Model parameters at step $t$ | $\theta_t \in \mathbb{R}^d$ |
| $\mathcal{L}(\theta)$ | Loss function | $L$-smooth (Assumption A.1) |
| $\bar{g}_t$ | Stochastic gradient estimate | Unbiased: $\mathbb{E}[\bar{g}_t] = \nabla\mathcal{L}(\theta_t)$ |
| | | Bounded Variance: $\mathbb{E}[\|\bar{g}_t - \nabla\mathcal{L}(\theta_t)\|^2] \leq \sigma_{sgd}^2$ |
| $P_t$ | Preconditioner Matrix | Symmetric Positive Definite ($S_{++}^d$) |
| | | Spectrum bounded: $\lambda_{min} I \preceq P_t \preceq \lambda_{max} I$ |
| $\xi_t$ | Privacy Noise Vector | $\xi_t \sim \mathcal{N}(0, \sigma^2 C^2 I)$ |
| | | $\mathbb{E}[\xi_t] = 0, \quad \mathbb{E}[\|\xi_t\|^2] = d\sigma^2 C^2$ |
| $\eta_t$ | Learning Rate | Decaying sequence, e.g., $\eta_t = 1/\sqrt{T}$ |

## A.2. Formal Assumptions

We restate the assumptions required for the analysis.

**Assumption A.1** ($L$-Smoothness). The objective function $\mathcal{L}$ is $L$-smooth, meaning its gradient is Lipschitz continuous. For all $x, y \in \mathbb{R}^d$:

$$\|\nabla\mathcal{L}(x) - \nabla\mathcal{L}(y)\| \leq L\|x - y\| \tag{6}$$

This implies the standard quadratic upper bound (Descent Lemma):

$$\mathcal{L}(y) \leq \mathcal{L}(x) + \langle \nabla\mathcal{L}(x), y - x \rangle + \frac{L}{2}\|y - x\|^2 \tag{7}$$

**Assumption A.2** (Independence of Noise). The privacy noise $\xi_t$ is drawn independently at each step and is independent of the stochastic gradient estimate $\bar{g}_t$. The preconditioner $P_t$ is computed using side information (public data or synthetic noise) and is therefore statistically independent of the current private batch gradients $\bar{g}_t$ and privacy noise $\xi_t$.

## A.3. Derivation

We proceed by analyzing the expected decrease in loss for a single iteration.

**Lemma A.3** (One-Step Descent Bound). *Under Assumptions A.1 and A.2, the expected loss at iteration $t + 1$, conditioned on the parameters $\theta_t$, satisfies:*

$$\mathbb{E}[\mathcal{L}(\theta_{t+1}) \mid \theta_t] \leq \mathcal{L}(\theta_t) - \eta_t \lambda_{min}\|\nabla\mathcal{L}(\theta_t)\|^2 + \frac{L\eta_t^2}{2}\lambda_{max}^2 \left(\|\nabla\mathcal{L}(\theta_t)\|^2 + \sigma_{sgd}^2\right) + \frac{L\eta_t^2}{2}d\sigma^2 C^2 \tag{8}$$

*Proof.* Let the update step be $\Delta_t = -\eta_t(P_t\bar{g}_t + \xi_t)$. Applying the $L$-smoothness inequality:

$$\mathcal{L}(\theta_{t+1}) \leq \mathcal{L}(\theta_t) + \langle \nabla\mathcal{L}(\theta_t), \Delta_t \rangle + \frac{L}{2}\|\Delta_t\|^2 \tag{9}$$

We take the expectation with respect to the stochasticity at step $t$ ($\bar{g}_t$ and $\xi_t$).

**1. Linear Term (Descent):**

$$\mathbb{E}[\langle \nabla\mathcal{L}(\theta_t), \Delta_t \rangle] = \mathbb{E}\left[\langle \nabla\mathcal{L}(\theta_t), -\eta_t(P_t\bar{g}_t + \xi_t) \rangle\right] \tag{10}$$

$$= -\eta_t\langle \nabla\mathcal{L}(\theta_t), P_t\mathbb{E}[\bar{g}_t] \rangle - \eta_t\langle \nabla\mathcal{L}(\theta_t), \mathbb{E}[\xi_t] \rangle \tag{11}$$

$$= -\eta_t\nabla\mathcal{L}(\theta_t)^\top P_t\nabla\mathcal{L}(\theta_t) \quad \text{(Since } \mathbb{E}[\bar{g}_t] = \nabla\mathcal{L}, \mathbb{E}[\xi_t] = 0) \tag{12}$$

We use the spectral property of positive definite matrices (Rayleigh quotient): for any vector $v$, $v^\top P_t v \geq \lambda_{min}(P_t)\|v\|^2$. Multiplying by $-\eta_t < 0$ reverses the inequality:

$$-\eta_t\nabla\mathcal{L}(\theta_t)^\top P_t\nabla\mathcal{L}(\theta_t) \leq -\eta_t\lambda_{min}\|\nabla\mathcal{L}(\theta_t)\|^2 \tag{13}$$

This term guarantees that the preconditioner provides a valid descent direction at least proportional to its smallest eigenvalue.

**2. Quadratic Term (Penalty):** We expand the squared norm of the update $\Delta_t = -\eta_t(P_t\bar{g}_t + \xi_t)$:

$$\mathbb{E}[\|\Delta_t\|^2] = \eta_t^2\mathbb{E}\left[\|P_t\bar{g}_t + \xi_t\|^2\right] \tag{14}$$

$$= \eta_t^2\left(\mathbb{E}[\|P_t\bar{g}_t\|^2] + \mathbb{E}[\|\xi_t\|^2] + 2\mathbb{E}[\langle P_t\bar{g}_t, \xi_t \rangle]\right) \tag{15}$$

We analyze the cross-term $\mathbb{E}[\langle P_t\bar{g}_t, \xi_t \rangle]$. By Assumption A.2, the privacy noise $\xi_t$ is statistically independent of the gradient estimate $\bar{g}_t$ and the fixed preconditioner $P_t$. Therefore, the expectation factorizes:

$$\mathbb{E}[\langle P_t\bar{g}_t, \xi_t \rangle] = \langle \mathbb{E}[P_t\bar{g}_t], \mathbb{E}[\xi_t] \rangle = \langle \mathbb{E}[P_t\bar{g}_t], \mathbf{0} \rangle = 0 \tag{16}$$

With the cross-term eliminated, we bound the remaining terms. For the gradient term, we use the spectral upper bound $\|P_t\|_2 \leq \lambda_{max}$ and the variance decomposition $\mathbb{E}[X^2] = (\mathbb{E}X)^2 + \text{Var}(X)$:

$$\mathbb{E}[\|P_t\bar{g}_t\|^2] \leq \lambda_{max}^2\mathbb{E}[\|\bar{g}_t\|^2] \tag{17}$$

$$= \lambda_{max}^2\left(\|\mathbb{E}[\bar{g}_t]\|^2 + \mathbb{E}[\|\bar{g}_t - \mathbb{E}[\bar{g}_t]\|^2]\right) \tag{18}$$

$$\leq \lambda_{max}^2\left(\|\nabla\mathcal{L}(\theta_t)\|^2 + \sigma_{sgd}^2\right) \tag{19}$$

For the noise term, using the definition of the multivariate Gaussian variance:

$$\mathbb{E}[\|\xi_t\|^2] = \text{Tr}(\text{Cov}(\xi_t)) + \|\mathbb{E}[\xi_t]\|^2 = \text{Tr}(\sigma^2 C^2 I_d) = d\sigma^2 C^2 \tag{20}$$

Substituting these back into the expansion yields the final quadratic bound.

$\square$

## A.4. Proof of Theorem 5.4

We proceed by summing the result of Lemma A.3 over iterations $t = 0, \dots, T-1$. Let $\eta_t = \eta$ be constant for simplicity. Rearranging Lemma A.3 to isolate the gradient norm:

$$\eta\lambda_{min}\|\nabla\mathcal{L}_t\|^2 \leq \mathcal{L}(\theta_t) - \mathbb{E}[\mathcal{L}(\theta_{t+1})] + \frac{L\eta^2}{2}M \tag{21}$$

where $M = \lambda_{max}^2(\sigma_{sgd}^2 + \|\nabla\mathcal{L}_t\|^2) + d\sigma^2 C^2$ is the noise/penalty constant.

Summing over $t$:

$$\eta\lambda_{min}\sum_{t=0}^{T-1}\mathbb{E}[\|\nabla\mathcal{L}_t\|^2] \leq \sum_{t=0}^{T-1}\left(\mathcal{L}(\theta_t) - \mathbb{E}[\mathcal{L}(\theta_{t+1})]\right) + \sum_{t=0}^{T-1}\frac{L\eta^2}{2}M \tag{22}$$

The first term on the RHS is a **telescoping sum**. The intermediate terms cancel out:

$$\sum_{t=0}^{T-1} (\mathcal{L}_t - \mathcal{L}_{t+1}) = \mathcal{L}_0 - \mathcal{L}_T \leq \mathcal{L}_0 - \mathcal{L}^* \tag{23}$$

where $\mathcal{L}^*$ is the global minimum. Substituting this back:

$$\eta \lambda_{min} \sum_{t=0}^{T-1} \mathbb{E}[\|\nabla \mathcal{L}_t\|^2] \leq (\mathcal{L}_0 - \mathcal{L}^*) + T \frac{L\eta^2}{2} M \tag{24}$$

Dividing both sides by $T\eta\lambda_{min}$, we obtain a bound on the *average* squared gradient norm:

$$\frac{1}{T} \sum_{t=0}^{T-1} \mathbb{E}[\|\nabla \mathcal{L}_t\|^2] \leq \frac{\mathcal{L}_0 - \mathcal{L}^*}{T\eta\lambda_{min}} + \frac{L\eta M}{2\lambda_{min}} \tag{25}$$

Since the minimum value in a sequence is always lower than or equal to the average ($\min_t x_t \leq \frac{1}{T}\sum x_t$), we have:

$$\min_{t \in [0, T-1]} \mathbb{E}[\|\nabla \mathcal{L}_t\|^2] \leq \underbrace{\frac{\mathcal{L}_0 - \mathcal{L}^*}{T\eta\lambda_{min}}}_{\text{(A) Optimization Error}} + \underbrace{\frac{L\eta M}{2\lambda_{min}}}_{\text{(B) Noise Variance}} \tag{26}$$

To minimize the RHS, we must balance term (A), which decreases with $\eta$, and term (B), which increases with $\eta$. We choose the decaying step size schedule $\eta = \frac{1}{\sqrt{T}}$. Substituting this into the bound:

**1. Optimization Term (A):**

$$\frac{\mathcal{L}_0 - \mathcal{L}^*}{T(1/\sqrt{T})\lambda_{min}} = \frac{\mathcal{L}_0 - \mathcal{L}^*}{\sqrt{T}\lambda_{min}} = \mathcal{O}\left(\frac{1}{\sqrt{T}}\right) \tag{27}$$

**2. Noise Variance Term (B):**

$$\frac{L(1/\sqrt{T})M}{2\lambda_{min}} = \frac{LM}{2\lambda_{min}\sqrt{T}} = \mathcal{O}\left(\frac{1}{\sqrt{T}}\right) \tag{28}$$

Since both terms decay at the same rate, the total convergence rate is $\mathcal{O}(1/\sqrt{T})$. This confirms that the algorithm converges to a stationary point despite the injected privacy noise. $\square$

## B. Theoretical Preconditioned Geometry

In this section, we rigorously establish that the preconditioning transformation derived from the KFAC factors successfully standardizes the geometry of the gradient distribution. Our goal is to prove that if the preconditioner accurately captures the second-moment matrix of the gradients, the transformed gradients become isotropic.

**Proposition B.1** (Isotropy of Preconditioned Gradients). *Let $g \in \mathbb{R}^d$ be a stochastic gradient vector with second moment matrix $F = \mathbb{E}[gg^\top]$. Consider the preconditioning transformation $P = F^{-1/2}$. The transformed gradient $\tilde{g} = Pg$ satisfies the isotropy condition $\mathbb{E}[\tilde{g}\tilde{g}^\top] = I_d$, and its expected squared norm is $\mathbb{E}[\|\tilde{g}\|^2] = d$.*

*Proof.* We first derive the covariance structure of the transformed gradient. By definition of the covariance matrix and substituting the linear transformation $\tilde{g} = F^{-1/2}g$, we have:

$$\text{Cov}(\tilde{g}) = \mathbb{E}[\tilde{g}\tilde{g}^\top] = \mathbb{E}\left[(F^{-1/2}g)(F^{-1/2}g)^\top\right]. \tag{29}$$

Using the property that the matrix transpose distributes over the product in reverse order, $(F^{-1/2}g)^\top = g^\top (F^{-1/2})^\top$. Since $F$ is a symmetric positive semi-definite matrix, its inverse square root $F^{-1/2}$ is also symmetric, implying $(F^{-1/2})^\top = F^{-1/2}$. Since $F^{-1/2}$ is a constant matrix with respect to the expectation over the data distribution, we can extract it from the expectation operator:

$$\mathbb{E}[\tilde{g}\tilde{g}^\top] = F^{-1/2}\mathbb{E}[gg^\top]F^{-1/2}. \tag{30}$$

Substituting the definition of the second moment matrix $F = \mathbb{E}[gg^\top]$ into the equation yields:

$$\text{Cov}(\tilde{g}) = F^{-1/2}FF^{-1/2} = F^{-1/2}(F^{1/2}F^{1/2})F^{-1/2} = I_d. \tag{31}$$

This confirms that the covariance of the preconditioned gradient is the identity matrix $I_d$, satisfying the definition of isotropy.

We now establish the scaling of the gradient norm. The expected squared $L_2$ norm is the trace of the second moment matrix. Using the linearity of the trace operator:

$$\mathbb{E}[\|\tilde{g}\|^2] = \mathbb{E}[\text{Tr}(\tilde{g}\tilde{g}^\top)] = \text{Tr}(\mathbb{E}[\tilde{g}\tilde{g}^\top]). \tag{32}$$

Substituting the result $\mathbb{E}[\tilde{g}\tilde{g}^\top] = I_d$ derived above:

$$\mathbb{E}[\|\tilde{g}\|^2] = \text{Tr}(I_d) = d. \tag{33}$$

Thus, preconditioning standardizes the gradient distribution such that the variance is uniformly distributed across all $d$ dimensions. $\square$

*Remark* B.2 (Choice of Preconditioning Transformation). It is important to distinguish our gradient preconditioning ($P = F^{-1/2}$) from the full Natural Gradient update ($P = F^{-1}$). If we were to use the full inverse, the covariance of the transformed gradient would be:

$$\text{Cov}(F^{-1}g) = F^{-1}FF^{-1} = F^{-1}. \tag{34}$$

This would invert the geometry rather than normalizing it to the identity sphere. By using the inverse square root $F^{-1/2}$, we ensure the preconditioned gradients are isotropic, so the spherical privacy noise matches the geometry of the gradients.

## C. Privacy Guarantees of Data-Free Preconditioning

A central claim of our work is that the introduction of the preconditioner $P_t$ does not degrade the privacy guarantees of the standard Gaussian mechanism. This relies on the independence of $P_t$ from the private dataset $\mathcal{D}$. Here, we formally prove that our mechanism satisfies differential privacy by demonstrating that the sensitivity of the update step remains bounded by the clipping norm $C$, regardless of the magnitude of the preconditioner.

**Lemma C.1** (Data-Independence of Synthetic Preconditioner). *Let $\mathcal{A}$ be the model architecture and $\mathcal{S} \sim \mathcal{D}_{noise}$ be a batch of synthetic noise drawn from a public distribution. The preconditioner computation map $\Psi(\mathcal{A}, \mathcal{S}) \to P_t$ satisfies Differential Privacy with $\epsilon = 0$ with respect to the private dataset $\mathcal{D}$.*

*Proof.* The function $\Psi$ depends exclusively on the model architecture definition $\mathcal{A}$ (which is public knowledge) and the synthetic batch $\mathcal{S}$ (which is generated computationally using a fixed random seed or public entropy). The private dataset $\mathcal{D}$ is not an input to $\Psi$. Therefore, for any two adjacent private datasets $\mathcal{D}, \mathcal{D}'$, the preconditioner remains identical: $P_t(\mathcal{D}) = P_t(\mathcal{D}')$. Consequently, the generation of $P_t$ incurs zero privacy loss. $\square$

**Theorem C.2** (Privacy of Preconditioned Updates). *Let $P_t$ be a fixed linear operator that is batch-independent (computed from synthetic noise and past privatized model states per Lemma C.1). The preconditioned update mechanism defined by:*

$$\mathcal{M}(\mathcal{B}) = \sum_{x_i \in \mathcal{B}} Clip_C(P_t \nabla \ell(x_i)) + \mathcal{N}(0, \sigma^2 C^2 I) \tag{35}$$

*satisfies the same $(\epsilon, \delta)$-Differential Privacy guarantees as the standard Gaussian mechanism with noise scale $\sigma$ and sensitivity $C$. Privacy accounting is performed on this noisy sum; subsequent averaging scales both signal and noise uniformly without affecting the privacy guarantee.*

*Proof.* The privacy guarantee of the Gaussian mechanism is determined entirely by the $L_2$-sensitivity of the query function it obfuscates. We define the query function $q$ over a batch $\mathcal{B}$ as the sum of clipped, preconditioned gradients:

$$q(\mathcal{B}) = \sum_{x_i \in \mathcal{B}} \text{Clip}_C(P_t \nabla \ell(x_i)). \tag{36}$$

We analyze the $L_2$-sensitivity $\Delta_2(q)$, which measures the maximum change in the query output resulting from replacing a single sample $x_k$ with $x'_k$ in the batch. Let $\tilde{g}_k = P_t \nabla \ell(x_k)$ be the preconditioned gradient. The sensitivity is given by:

$$\Delta_2(q) = \max_{\mathcal{B}, \mathcal{B}'} \|q(\mathcal{B}) - q(\mathcal{B}')\|_2 = \max_{x_k, x'_k} \|\text{Clip}_C(\tilde{g}_k) - \text{Clip}_C(\tilde{g}'_k)\|_2. \tag{37}$$

The clipping operator $\text{Clip}_C(v) = v \cdot \min(1, C/\|v\|_2)$ ensures that for any input vector $v$, the magnitude of the output is strictly bounded such that $\|\text{Clip}_C(v)\|_2 \leq C$. By the triangle inequality, the maximum difference between two vectors, each with a norm bounded by $C$, is $2C$ (in the worst case of opposite directions). However, in the context of DP-SGD, sensitivity is defined regarding the *addition* or *removal* of a single record. For the replacement definition standard in batch-level accounting, the sensitivity is bounded by the maximum contribution of a single record:

$$\sup_x \|\text{Clip}_C(P_t \nabla \ell(x))\|_2 \leq C. \tag{38}$$

The magnitude of the preconditioner $P_t$ effectively scales the raw gradient $\nabla \ell(x_k)$, potentially resulting in a vector $\tilde{g}_k$ with a very large norm. However, the clipping operation is applied *after* the matrix multiplication. The projection $\text{Clip}_C(\cdot)$ projects any vector, regardless of how large $\|P_t \nabla \ell\|_2$ becomes, onto the $L_2$ ball of radius $C$.

Therefore, the global sensitivity is bounded by $\Delta_2(q) \leq C$. Since the noise added is $\mathcal{N}(0, \sigma^2 C^2 I)$, the signal-to-noise ratio is governed strictly by $\sigma$, preserving the standard privacy analysis. $\square$

# D. Computational Complexity and Resource Efficiency

We analyze the computational trade-offs of DP-KFC. We distinguish between the *arithmetic complexity* (FLOPs per step) and the *sample complexity* (steps to convergence), arguing that in privacy-constrained settings, the latter is the dominant factor.

## D.1. Arithmetic Complexity Analysis

Let a neural network layer $l$ have input dimension $d_{in}$ and output dimension $d_{out}$. The parameter matrix is $W \in \mathbb{R}^{d_{out} \times d_{in}}$.

**Standard DP-SGD Cost.** The dominant cost in standard DP-SGD is the per-sample gradient computation required for clipping. For a batch size $B$, this involves computing the outer product of error signals $\delta \in \mathbb{R}^{d_{out}}$ and activations $a \in \mathbb{R}^{d_{in}}$:

$$g^{(i)} = \delta^{(i)}(a^{(i)})^\top. \tag{39}$$

The cost is $\mathcal{O}(B \cdot d_{in}d_{out})$ per layer.

**DP-KFC Cost (Per-Step).** Our method applies the preconditioning transformation $\tilde{g} = U_G g U_A^\top$. Naively implementing this as matrix-matrix multiplications would be prohibitively expensive ($\mathcal{O}(d_{out}^2 d_{in} + d_{in}^2 d_{out})$). However, exploiting the rank-1 structure of the gradient $g^{(i)}$, we apply the transformation directly to the activation and error vectors:

$$\tilde{g}^{(i)} = (U_G \delta^{(i)})(U_A a^{(i)})^\top. \tag{40}$$

This reduces the operation to two matrix-vector products per sample. The additional cost is $\mathcal{O}(B(d_{in}^2 + d_{out}^2))$. Since $d_{in}^2 + d_{out}^2$ is typically comparable to $d_{in}d_{out}$ (the cost of the backward pass), the total FLOPs per iteration increase by a factor of $2\times$ to $3\times$ compared to DP-SGD, preserving the linear scaling with batch size $B$. Empirically, on our benchmarks (NVIDIA A100) this manifests as a $\approx 2.2\times$ reduction in training sample throughput relative to DP-SGD, consistent with the analytical estimate.

**Preconditioner Update Cost (Amortized).** The eigendecomposition of the covariance matrices requires $\mathcal{O}(d_{in}^3 + d_{out}^3)$ operations. Since this is performed only every $T_{freq}$ steps using synthetic data, the amortized complexity is negligible for standard frequencies (e.g., $T_{freq} = 50$):

$$\mathcal{C}_{amortized} = \frac{1}{T_{freq}} \sum_{l=1}^{L} \left[ \underbrace{M(d_{in}^2 + d_{out}^2)}_{\text{Covariance Est.}} + \underbrace{(d_{in}^3 + d_{out}^3)}_{\text{Eigen-Decomp.}} \right] \approx 0. \tag{41}$$

## D.2. The Privacy-Compute Trade-off

Critiques of second-order methods often focus on wall-clock time. We argue that this metric is ill-posed for Differential Privacy, where the optimization horizon is bounded by the privacy budget $\epsilon$, not computational resources.

**The Privacy Wall Hypothesis.** In non-private learning, one can trade time for accuracy: given infinite compute $T \to \infty$, the error approaches zero. In private learning, the noise scale $\sigma$ required for a fixed $\epsilon$ grows with $\sqrt{T}$ (using RDP accounting). This creates a *Privacy Wall*:

- **Non-Private Constraint:** $\min_\theta \mathcal{L}(\theta)$    s.t.    WallClockTime $\leq T_{limit}$

- **Private Constraint:** $\min_\theta \mathcal{L}(\theta)$    s.t.    PrivacyLoss$(T) \leq \epsilon_{budget}$

Because the number of steps $T$ is strictly capped by $\epsilon_{budget}$, the only mechanism to improve final utility is to increase the *information gain per gradient step*. As summarized in Table 4, DP-KFC invests renewable resources (FLOPs) to conserve non-renewable resources (Privacy Budget). By improving the convergence rate per step $\kappa(\mathcal{L})$, we maximize the utility of the limited queries allowed before hitting the Privacy Wall.

# E. Synthetic Structural Probing

In this section, we provide the implementation details for generating the synthetic structural probes used to estimate the KFAC factors $A$ and $G$ without private data.

*Table 4.* **Resource Consumption Analysis.** Comparison of costs for a fixed privacy budget $\epsilon$. $T_{conv}$ denotes steps to convergence. DP-KFC trades linear arithmetic overhead for a reduction in the required optimization steps.

| Method | FLOPs / Step | Memory | Steps to Conv. | Limiting Factor |
|---|---|---|---|---|
| DP-SGD | $1\times$ | $\mathcal{O}(d)$ | High ($T_{sgd}$) | Privacy Budget |
| Full KFAC | $\mathcal{O}(d^2)$ | $\mathcal{O}(d^2)$ | Low ($T_{kfac}$) | Compute / Memory |
| **DP-KFC (Ours)** | $\mathbf{2-3\times}$ | $\mathcal{O}(\mathbf{d})$ | **Medium ($T_{small}$)** | **Compute** |

### E.1. Continuous Domain: Pink Noise Generation

For Convolutional Neural Networks (CNNs) and Vision Transformers (ViTs) operating on continuous signals, we generate **Pink Noise** (also known as $1/f$ noise). Natural images exhibit a power-law spectral density $S(f) \propto 1/f^\alpha$, typically with $\alpha \in [1,2]$. Probing with white noise ($\alpha = 0$) creates a flat spectrum that disproportionately amplifies high-frequency spatial modes which are often dampened by the inductive bias of visual architectures (e.g., pooling layers, blur kernels).

To construct a spatially correlated probe that matches the $1/f$ statistics of natural images, we operate in the frequency domain. Let $H, W$ be the spatial dimensions and $C$ be the number of channels.

**Generation Procedure:**

1. **White Noise Source:** We sample a standard Gaussian tensor in the frequency domain. To ensure the resulting spatial signal is real-valued, we sample complex values $Z \in \mathbb{C}^{H \times W}$ such that the Hermitian symmetry $Z(-u) = \overline{Z(u)}$ is preserved (or simply sample in the spatial domain and apply FFT).

2. **Spectral Modulation:** We compute the frequency grid $\mathbf{u} \in \mathbb{R}^{H \times W}$, where $\mathbf{u}_{ij}$ represents the Euclidean distance from the DC component (frequency zero). We scale the amplitude of the noise by the inverse frequency:

$$\tilde{Z}_{\mathbf{u}} = Z_{\mathbf{u}} \cdot \frac{1}{\|\mathbf{u}\|_2^{\alpha/2} + \epsilon} \tag{42}$$

   where $\epsilon$ prevents division by zero at the DC component.

3. **Inverse Transform:** We apply the Inverse Fast Fourier Transform (IFFT) to obtain the spatial signal $x_{\text{pink}} = \text{Re}(\mathcal{F}^{-1}(\tilde{Z}))$.

4. **Normalization:** Since KFAC factor estimation depends on the relative variance scale, we normalize the resulting batch to zero mean and unit variance to match the standard initialization statistics of the network weights.

Algorithm 3 details the implementation.

### E.2. Discrete Domain: Structural Token Noise

For Large Language Models (LLMs) and Transformers, "white noise" in the embedding space is insufficient because it creates vectors that do not lie on the discrete manifold of valid token embeddings. Furthermore, Transformers possess strong architectural constraints—specifically **padding masks** and **special token anchors** (e.g., [CLS], [SEP])—that dictate the sparsity pattern of the attention mechanism and the backward error flow.

To recover the geometry of the self-attention blocks ($A_{l-1}$) and the feed-forward networks, we generate **Structural Token Noise**. Instead of sampling continuous vectors, we sample integer sequences that respect the model's structural requirements.

**Generation Procedure:**

1. **Variable Length Sampling:** To capture the Fisher dynamics across different context windows, we do not fix the sequence length. For each synthetic batch, we sample a random active length $L_{\text{act}} \sim \mathcal{U}(L_{\min}, L_{\max})$.

2. **Vocabulary Sampling:** For the active positions, we sample integers uniformly from the model's vocabulary $v \sim \mathcal{U}(0, V_{\text{vocab}})$. This ensures that when projected through the model's embedding layer, the inputs activate the lookup table with the correct variance $q^0 = \text{Var}(W_{\text{embed}})$.

3. **Structural Anchoring:** We enforce the insertion of special tokens (e.g., [CLS] at index 0) required by the architecture.

---

**Algorithm 3** Synthetic Pink Noise Generator ($1/f^\alpha$)

---

**Input:** Batch size $B$, Channels $C$, Height $H$, Width $W$, Decay $\alpha$ (default 1.0).
**Output:** Synthetic Batch $X \in \mathbb{R}^{B \times C \times H \times W}$.
 1: **1. Frequency Grid:**
 2: Create coordinate grid $(u, v)$ centered at zero.
 3: Compute radius $r = \sqrt{u^2 + v^2}$.
 4: Define Filter $S = 1/(r^\alpha + \epsilon)$.
 5: **2. Generate White Noise:**
 6: $X_{\text{white}} \sim \mathcal{N}(0, 1)$ of shape $(B, C, H, W)$.
 7: $Z_{\text{white}} = \text{FFT2D}(X_{\text{white}})$.
 8: **3. Modulate Spectrum:**
 9: $Z_{\text{pink}} = Z_{\text{white}} \odot S$ {Element-wise scaling}
10: **4. Spatial Reconstruction:**
11: $X_{\text{pink}} = \text{Real}(\text{IFFT2D}(Z_{\text{pink}}))$.
12: **5. Normalize:**
13: $X \leftarrow \frac{X_{\text{pink}} - \text{mean}(X_{\text{pink}})}{\text{std}(X_{\text{pink}})}$ {Match standard init variance}
14:
15: **return** $X$

---

4. **Mask Construction:** We explicitly construct the binary `attention_mask`. Positions $i > L_{\text{act}}$ are set to 0 (padded). This is critical for KFAC, as it ensures the estimated covariance matrices correctly reflect the sparsity induced by the masking operation in the backward pass.

Algorithm 4 details the implementation.

---

**Algorithm 4** Structural Token Noise Generator

---

**Input:** Batch size $B$, Max Length $L_{\max}$, Vocab Size $V$, Special Tokens (CLS, PAD).
**Output:** Input IDs $X \in \mathbb{Z}^{B \times L}$, Mask $M \in \{0, 1\}^{B \times L}$.
 1: Initialize $X \leftarrow \text{fill}(B, L, \text{PAD})$.
 2: Initialize $M \leftarrow \text{zeros}(B, L)$.
 3: **for** sample $i = 1 \ldots B$ **do**
 4:     Sample length $\ell_i \sim \mathcal{U}(1, L_{\max})$.
 5:     // 1. Generate Random Payload
 6:     Sample tokens $t \sim \mathcal{U}(0, V)$ of size $\ell_i$.
 7:     $X[i, 0 : \ell_i] \leftarrow t$.
 8:     // 2. Enforce Structural Anchors
 9:     $X[i, 0] \leftarrow \text{CLS}$.
10:     **if** uses_separator **then**
11:         $X[i, \ell_i] \leftarrow \text{SEP}$.
12:     **end if**
13:     // 3. Generate Valid Mask
14:     $M[i, 0 : \ell_i] \leftarrow 1$.
15:     // Remaining positions stays 0 (PAD)
16: **end for**
17:
18: **return** $X, M$ {Fed into model to get embeddings}

---

### E.3. KFAC for Modern Architectures

The Kronecker preconditioning $\tilde{g}_l = U_{G,l} \cdot g_l \cdot U_{A,l}$ assumes the gradient $g_l \in \mathbb{R}^{d_{out} \times d_{in}}$ is a matrix, which is natural for fully-connected layers. Here we detail how KFAC extends to convolutional and attention layers.

**Convolutional Layers.** For a Conv2d layer with kernel size $k \times k$, input channels $C_{in}$, and output channels $C_{out}$, the gradient tensor has shape $(C_{out}, C_{in}, k, k)$. We apply the im2col transformation (Martens & Grosse, 2015): the input

activation tensor $X \in \mathbb{R}^{B \times C_{in} \times H \times W}$ is unfolded into patches $\tilde{X} \in \mathbb{R}^{n \times (C_{in} \cdot k^2)}$, where $n = B \cdot H_{out} \cdot W_{out}$ treats each spatial location as an independent sample. The activation covariance becomes:

$$\hat{A} = \frac{1}{n} \tilde{X}^\top \tilde{X} \in \mathbb{R}^{(C_{in} \cdot k^2) \times (C_{in} \cdot k^2)} \tag{43}$$

Similarly, the gradient tensor is reshaped to $\tilde{G} \in \mathbb{R}^{n \times C_{out}}$, yielding $\hat{G} = \frac{1}{n} \tilde{G}^\top \tilde{G}$. This reduces convolutional preconditioning to the matrix case.

**Transformer Architectures.** Vision Transformers (ViT) and language models (BERT) consist primarily of Linear layers: attention projections ($Q$, $K$, $V$, output) and MLP blocks. These apply standard KFAC directly. The weight-sharing across sequence positions (each token uses the same projection weights) is handled via the K-FAC-reduce formulation (Eschenhagen et al., 2023): activations from all sequence positions are pooled together when computing covariances. For an activation tensor $A \in \mathbb{R}^{B \times T \times d}$ (batch, sequence, features), we reshape to $(B \cdot T) \times d$ and compute:

$$\hat{A} = \frac{1}{B \cdot T} A_{flat}^\top A_{flat} \tag{44}$$

This averaging over the weight-sharing dimension is computationally efficient and empirically equivalent to the more expensive K-FAC-expand variant that maintains separate covariances per position.

**Memory Considerations.** For architectures with large convolutional kernels (e.g., ConvNeXt with $7 \times 7$ depthwise convolutions), the covariance matrix size is $O((C_{in} \cdot k^2)^2)$, which can be prohibitive. In such cases, we apply KFAC only to the Linear layers (classifier head, MLP blocks) while leaving large convolutions unpreconditioned. This hybrid approach captures the dominant geometry while avoiding excessive memory costs.

## F. Spectral Analysis and Theoretical Justification

In this section, we provide the theoretical formulations for the eigenspectrum alignment observed in Section 6.2. We connect results from Mean Field Theory (MFT) and Random Matrix Theory (RMT) to justify why synthetic noise probing recovers the essential eigenspectrum of the preconditioner matrix required for effective preconditioning. We then detail the empirical methodologies used to validate this theory.

### F.1. Theoretical Foundations

Our method relies on the insight that the spectral decay of layer-wise covariance matrices is an asymptotic property of the architecture depth and non-linearity, rather than a unique feature of the data manifold.

#### F.1.1. MEAN FIELD VARIANCE DYNAMICS

Consider a deep neural network with $L$ layers in the infinite-width limit. Let $q_{ab}^l$ denote the covariance between the pre-activations of two inputs $x_a, x_b$ at layer $l$. The propagation of this covariance is governed by a deterministic recurrence map $\mathcal{V}$ (Schoenholz et al., 2016):

$$q_{ab}^l = \sigma_w^2 \mathbb{E}_{z_1, z_2 \sim \mathcal{N}(0, \Sigma_{l-1})}[\phi(z_1)\phi(z_2)] + \sigma_b^2 \tag{45}$$

A key result in signal propagation theory is the existence of a stable fixed point $q^*$ for the variance map $\mathcal{V}$ under standard initializations.

**Proposition F.1** (Variance Contraction). *Assume a bounded activation function $\phi$ and initialization parameters in the ordered phase. For any two input distributions $\mathcal{D}_{priv}$ and $\mathcal{D}_{syn}$, the layer-wise variances $q^l$ converge exponentially to the same fixed point $q^*$ as depth $L \to \infty$:*

$$\lim_{l \to \infty} |q^l(\mathcal{D}_{priv}) - q^l(\mathcal{D}_{syn})| = 0 \tag{46}$$

*Proof Sketch.* The map $q^l = \mathcal{V}(q^{l-1})$ acts as a contraction mapping on the domain of variances. Initial differences in signal magnitude $q^0$ between private data and noise are attenuated exponentially with depth, fixing the signal scale to the architectural fixed point $q^*$.

F.1.2. SPECTRAL SHAPE

The KFAC preconditioner relies on the spectra of the factors $A_{l-1} = \mathbb{E}[aa^\top]$ and $G_l = \mathbb{E}[gg^\top]$. According to Pennington & Bahri (2017), the limiting spectral density $\rho(\lambda)$ of these matrices follows a power-law decay determined by the architectural parameters.

**Theorem F.2** (Spectral Scaling Invariance). *Let $\lambda_k(A_l)$ be the $k$-th eigenvalue of the covariance at layer $l$, and assume the synthetic probe induces a non-degenerate first-layer signal variance $q_{syn}^0 > 0$. In the large depth limit, the spectral density converges to a form independent of the fine-grained input structure. Specifically, for private data and synthetic noise:*

$$\frac{\lambda_k(A_l^{priv})}{\lambda_k(A_l^{syn})} \approx \alpha \quad \forall k \tag{47}$$

*where $\alpha > 0$ is a scalar scaling factor derived from the initial variance ratio.*

*Proof Sketch.* By Proposition F.1, the scalar variance state $q^l$ contracts to the architectural fixed point $q^*$ regardless of the input distribution, so the *shape* of the limiting covariance $A_l$ (its eigenvector basis and the relative magnitudes $\lambda_k/\lambda_1$) is governed by the recurrence map $\mathcal{V}$ and its Jacobian, not by the input statistics; only an overall multiplicative constant $\alpha$, set by how the finite-depth trajectory enters the basin around $q^*$, depends on $q_{priv}^0/q_{syn}^0$. The same argument applied to the dual (backward) recurrence governs $G_l$. Composing, the layer-wise Fisher block satisfies $F_l^{priv} \approx \alpha_l F_l^{syn}$ for a scalar $\alpha_l$, hence $(\tilde{F}_l^{-1/2}) F_l^{priv} (\tilde{F}_l^{-1/2}) \approx \alpha_l I$.

Empirically, we observe this scaling in Fig. 6 where curves maintain consistent relative ordering. Since our preconditioning transformation $U_A = Q\Lambda^{-1/2}Q^\top$ normalizes based on the relative spread of eigenvalues, the per-layer scalar factors $\alpha_l$ are absorbed into the global clipping norm and learning rate. Thus, the noise-derived preconditioner correctly normalizes the geometry of the private data. We caution that this is an asymptotic (large-depth) statement: the directional ($Q$) alignment it predicts holds well in early layers but degrades in deeper layers, where eigenvector directions become data- (and label-) dependent (Appendix G); the *scale* alignment, which is what the clipping norm relies on, remains accurate throughout (Figs. 3, 7).

## F.2. Empirical Verification Methodology

We validate these theoretical claims using two complementary spectral analysis techniques.

F.2.1. EIGENVALUE RECONSTRUCTION VIA KFAC FACTORS

Direct computation of the full Hessian spectrum is intractable. We approximate the spectrum of the Fisher block $F_l \approx A_{l-1} \otimes G_l$ analytically. Let the eigendecomposition of the empirical factors be $\hat{A}_{l-1} = Q_A \Lambda_A Q_A^\top$ and $\hat{G}_l = Q_G \Lambda_G Q_G^\top$. The spectrum of the approximated Fisher matrix is given by:

$$\Lambda_{F_l} = \text{sort}(\{\lambda_i^A \cdot \lambda_j^G \mid 1 \leq i \leq d_{in}, 1 \leq j \leq d_{out}\}) \tag{48}$$

This reconstruction allows us to visualize the spectral alignment across CNNs, MLPs and attention based architectures shown in Fig. 6.

F.2.2. FULL DENSITY ESTIMATION VIA STOCHASTIC LANCZOS QUADRATURE

To verify that the alignment holds beyond the KFAC approximation, we employ Stochastic Lanczos Quadrature (SLQ) to estimate the spectral density of the full Hessian $H$. SLQ approximates the density by computing moments $\mu_k = v^\top H^k v$ via random probes $v \sim \mathcal{N}(0, I)$.

The Hessian spectrum decomposes into a bulk (architecture-determined eigenvalues) and outliers (data-specific constraints) (Sagun et al., 2018). Our noise probing method effectively captures the "bulk" geometry, which spans the orders of magnitude responsible for ill-conditioning. Fig. 7 confirms that the support of the noise-estimated spectrum matches the true private spectrum across different architectures.

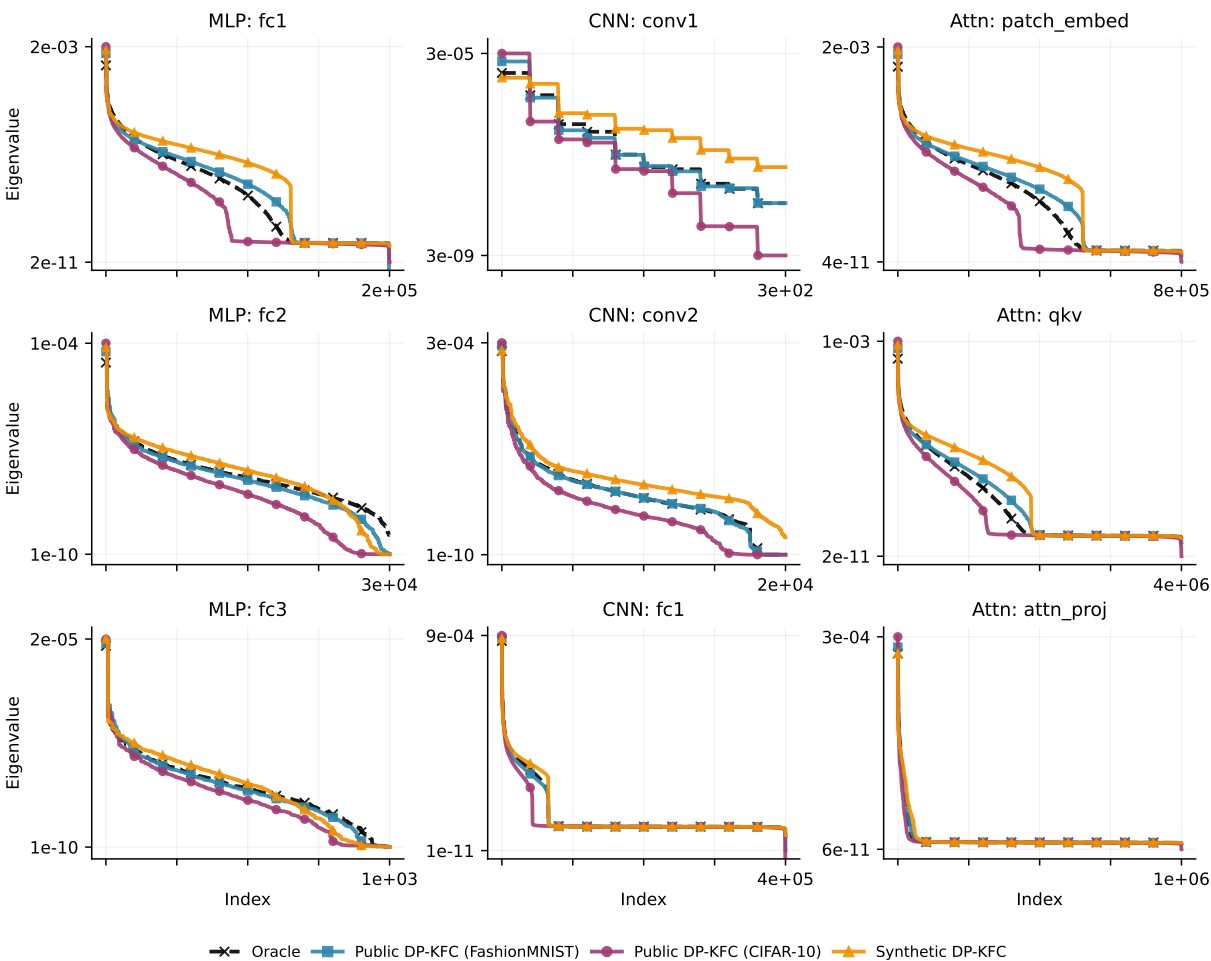

*Figure 6.* **KFAC Eigenspectrum Across Architectures and Data Sources.** Comparison of KFAC eigenvalue spectra for (a) MLP, (b) CNN, and (c) Attention architectures. Each panel shows the sorted eigenvalues of the Kronecker-factored Fisher approximation computed using: Oracle (private MNIST data), Public DP-KFC with FashionMNIST or CIFAR-10, and Synthetic DP-KFC (pink noise). The synthetic preconditioner closely tracks the Oracle spectrum shape across all architectures, confirming that network structure, not training data, primarily determines the eigenvalue distribution. Domain-matched public data (FashionMNIST) provides marginal improvement over synthetic noise.

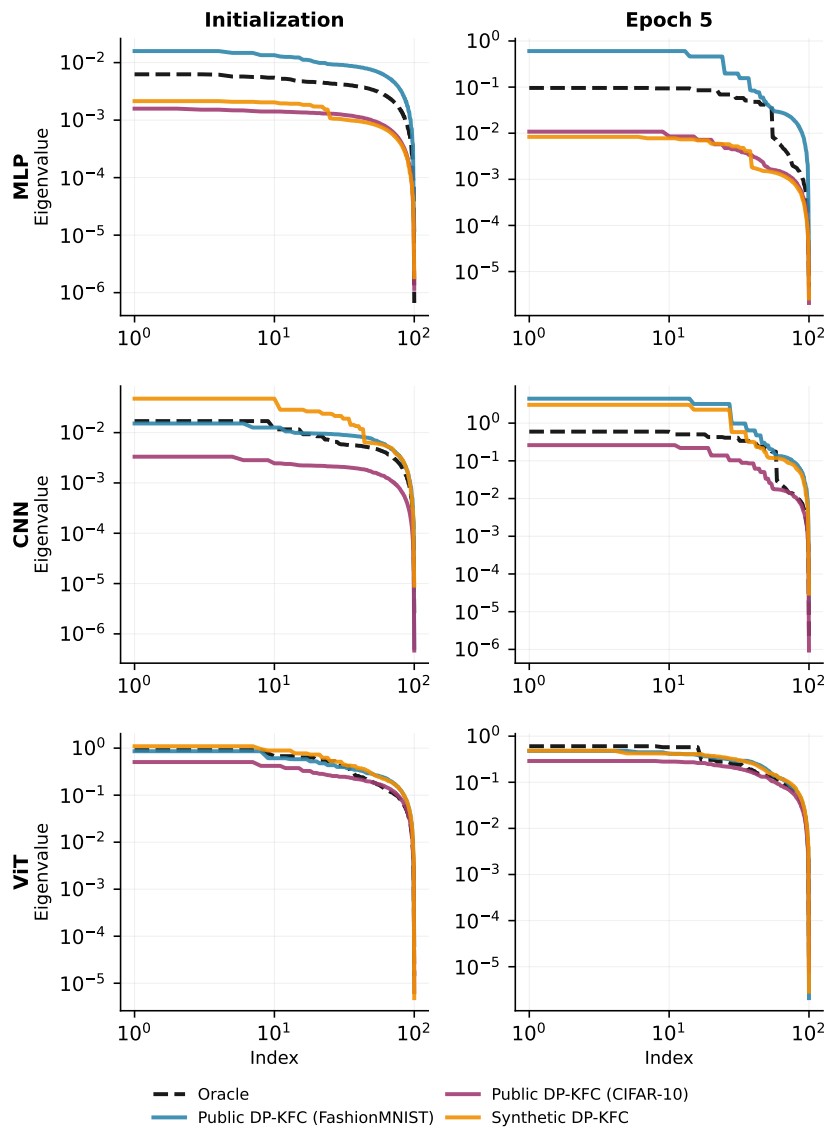

*Figure 7.* **Eigenvalue Decay via Stochastic Lanczos Quadrature.** Sorted eigenvalue magnitudes at initialization (left) and after training (right) for MLP, CNN, and ViT architectures. The Synthetic DP-KFC estimator (orange) tracks the decay profile of the Oracle spectrum (black, dashed) across all architectures, confirming that network structure primarily determines the eigenvalue distribution. Public data sources (blue, purple) show similar alignment.

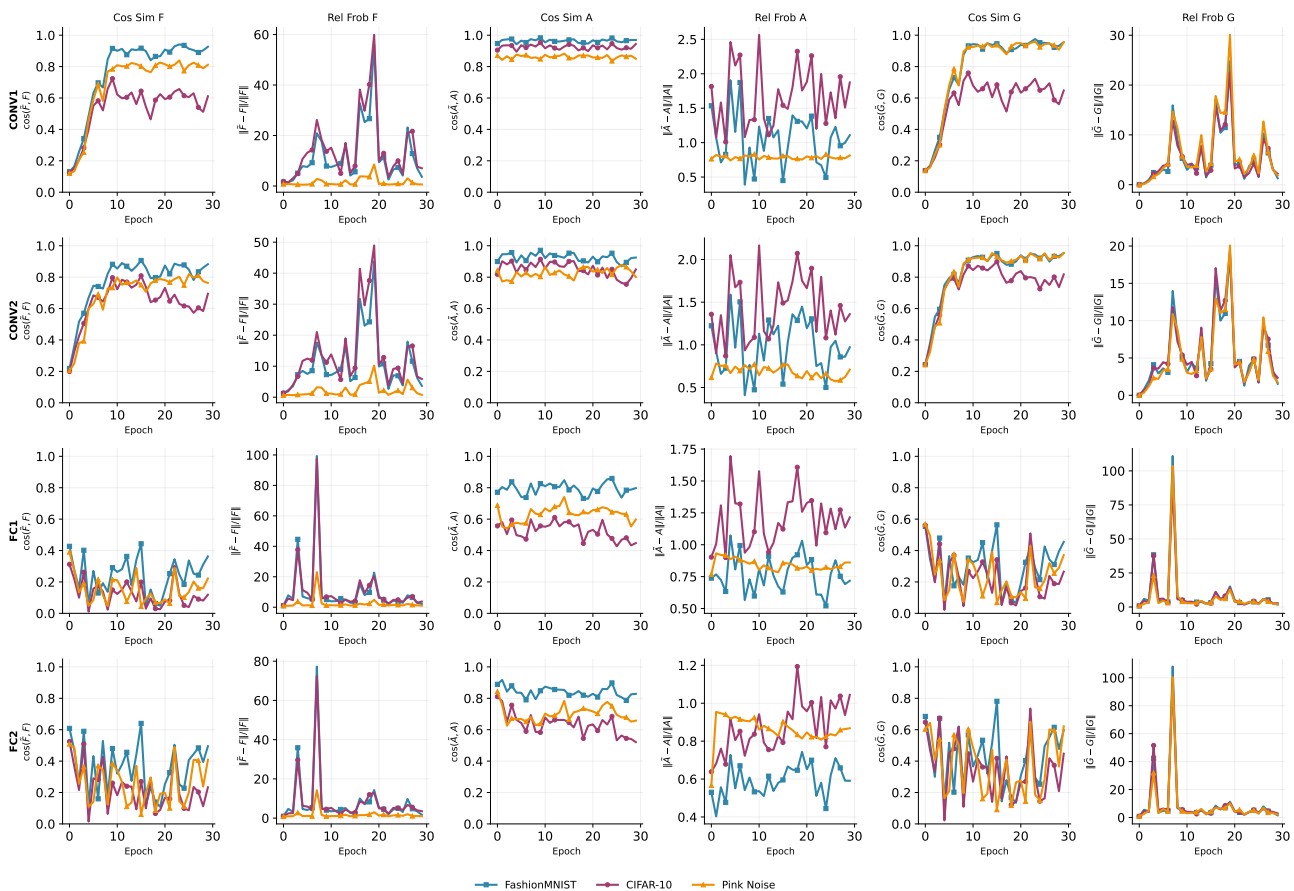

*Figure 8.* **Component Stability Analysis.** Decomposition of KFAC estimation error into Activation ($A$) and Gradient ($G$) factors. While backward error scale ($G$) is similar across methods, public data proxies cause catastrophic instability in the forward activation scale ($A$) due to feature specialization. Synthetic DP-KFC (Green) effectively stabilizes $A$, driving overall robustness.

# G. Covariance Factor Decomposition

We analyze the approximation quality of the KFAC factors, decomposing the Fisher Information Matrix into activation covariance $A_{l-1} = \mathbb{E}[a_{l-1}a_{l-1}^\top]$ and gradient covariance $G_l = \mathbb{E}[\delta_l \delta_l^\top]$. We track the alignment of our data-free estimator against private oracle statistics throughout training on a 4-layer CNN (MNIST, $\epsilon = 1.0$), comparing it to a matched public proxy (FashionMNIST) and a mismatched proxy (CIFAR-10).

## G.1. Metrics and Methodology

To characterize the preconditioner estimation quality, we utilize two complementary metrics. First, **Cosine Similarity** measures directional alignment: $\text{CosSim}(C^*, \hat{C}) = \text{Tr}(C^{*\top}\hat{C})/(\|C^*\|_F \|\hat{C}\|_F)$. High similarity indicates that the estimator correctly identifies the principal eigenvector directions of the preconditioner matrix. Second, **Relative Frobenius Error** captures scale mismatch: $\text{RelFrob}(C^*, \hat{C}) = \|C^* - \hat{C}\|_F / \|C^*\|_F$. Since preconditioning inverses these matrices, scale errors directly impact step size. For the full Fisher block, similarity is multiplicative: $\text{CosSim}(F^*, \hat{F}) \approx \text{CosSim}(A^*, \hat{A}) \cdot \text{CosSim}(G^*, \hat{G})$, implying that misalignment in either factor degrades the overall update.

## G.2. Results and Interpretation

Fig. 8 details the evolution of covariance alignment across network depth.

**Activation Covariance ($A$): The Stability of Noise.** We observe a fundamental dichotomy in forward pass stability. Mismatched public proxies (CIFAR-10) exhibit high variance in activation statistics, with Frobenius errors spiking by orders of magnitude. We attribute this to *Feature Specialization*: coherent features in natural images (e.g., textures) disproportionately activate filters optimized to the private manifold, causing activation explosions. In contrast, the Pink

Noise estimator remains remarkably stable (error $< 0.5$). The phase incoherence of pink noise ensures energy is distributed across frequencies ($1/f$) without triggering semantic feature detectors, effectively probing the architecture to estimate scale of the activations without saturation.

**Gradient Covariance ($G$): Architectural Dominance.** The backward error signal is largely method-agnostic. All methods exhibit synchronized error spikes correlated with hard batches in the private loop, as no data-free method can predict label-dependent supervision. However, despite these fluctuations, Pink Noise matches the directional alignment of the semantically related FashionMNIST proxy ($> 0.8$ Cosine Sim). This validates that the *architectural scaling* of the backward pass, governed by layer-wise Jacobians, dominates over task-specific label structure.

**Decoupling in Deep Layers.** The combined metrics reveal that Fisher estimation decouples with depth. In early layers (conv1, conv2), Pink Noise maintains high directional alignment ($> 0.8$), proving that spatial correlation structure ($1/f$) suffices to recover feature extraction geometry in image tasks. In deep layers (fc1, fc2), directional alignment degrades across all methods ($< 0.5$), reflecting the dependency on private labels. However, Pink Noise strictly improves the *scale stability*. While public proxies suffer from high variance due to input mismatch, noise-derived scaling remains robust. This confirms that while the *eigenvector directions* of the Fisher matrix in deep layers are data-dependent, the *eigenvalue magnitudes* are intrinsic architectural properties recoverable via synthetic probing.

These findings suggest that synthetic noise is strictly preferable to mismatched public data. It avoids negative transfer in the forward pass by preventing feature collisions while maintaining robust scaling in the backward pass. Even when public data is available, synthetic noise provides comparable alignment with greater stability against batch-specific outliers.

### G.3. Extension to Attention-Based Architectures

We repeat the covariance tracking analysis on a simple Vision Transformer (2-block, 4-head, embed_dim=64) trained on MNIST ($\epsilon = 1.0$, 30 epochs). To compress the 14 linear layers into a readable comparison, we group them into three categories: Attention (8 Q/K/V/output projections), MLP (4 feed-forward layers), and Embedding/Head (patch projection and classifier).

Fig. 9 shows the evolution of Fisher factor alignment throughout training. The source ranking reproduces the CNN pattern (Fig. 3): domain-matched FashionMNIST achieves the highest directional alignment, Synthetic DP-KFC provides a robust intermediate, and domain-mismatched CIFAR-10 performs worst on both metrics across all layer groups. Decomposing into activation and gradient factors confirms the same mechanism: gradient covariance alignment ($\cos_G$) is similar across all three sources ($\sim 0.4$–$0.5$), indicating that backward-pass geometry is architecture-dominated even in multi-head attention projections, while activation covariance ($\cos_A$) exhibits the largest gap between sources.

Absolute alignment is lower than in the CNN ($\cos_F$ of 0.3–0.4 vs. 0.6–0.8 in early convolutional layers), reflecting the increased modeling challenge of global self-attention compared to local convolutions. The Embedding/Head group shows elevated Frobenius error for all methods, consistent with the depth-dependent degradation observed in fully-connected layers of the CNN.

**LayerNorm and scale stability.** One notable difference from the CNN analysis is the relative Frobenius error. In the CNN, synthetic noise achieved strictly lower scale error than all public proxies, because mismatched natural images triggered feature-specific activation magnitudes that inflated $\|A_l\|_F$ (Section G). In the ViT, the matched proxy (FashionMNIST) achieves comparable or slightly lower Frobenius error than synthetic noise. We attribute this to LayerNorm, which is applied before every attention and MLP block. Since $\text{LN}(a) = \gamma \odot (a - \mu_a)/\sigma_a + \beta$, the pre-activation statistics are renormalized to zero mean and unit variance regardless of the input distribution. This constrains the Frobenius norm of the activation covariance: $\|A_l\|_F \approx \|\mathbb{E}[\hat{a}_l \hat{a}_l^\top]\|_F$ where $\hat{a}_l$ has bounded second moments by construction. The normalization suppresses the scale explosions that penalize mismatched public data in the CNN, compressing Frobenius errors across sources. As a result, the scale advantage that synthetic noise derives from phase-incoherent activations is diminished, because LayerNorm already prevents any single source from producing anomalous activation magnitudes. The directional ranking, however, is unaffected by this normalization and remains consistent with the CNN findings.

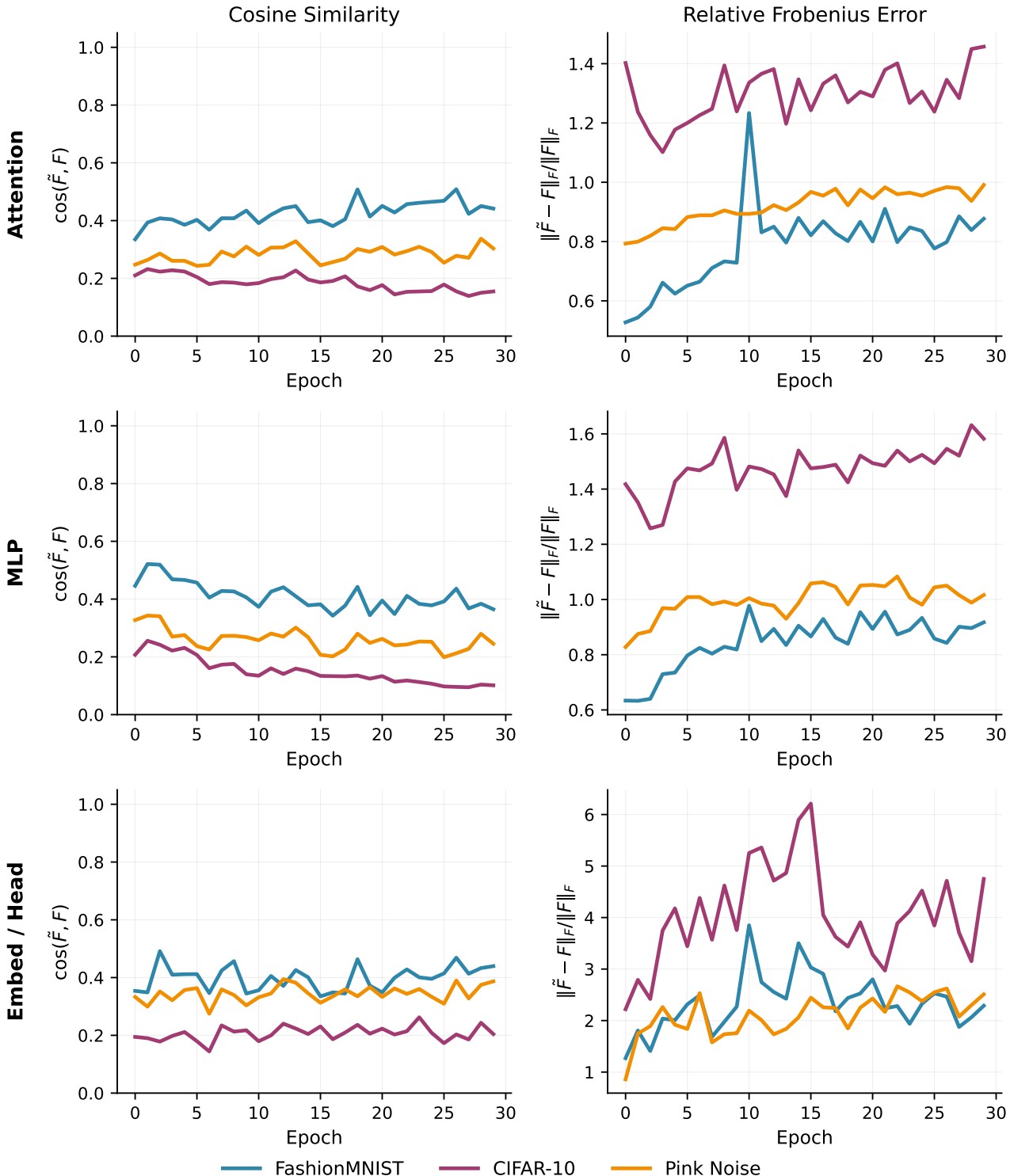

*Figure 9.* **Covariance Tracking in a Vision Transformer.** Cosine similarity (left) and relative Frobenius error (right) of the combined Fisher factor, averaged over layers within each group: Attention projections (top), MLP blocks (middle), and Embedding/Head (bottom). The source ranking is consistent with the CNN analysis (Fig. 3): domain-matched FashionMNIST (blue) achieves the highest alignment, Synthetic DP-KFC (orange) provides a robust intermediate, and mismatched CIFAR-10 (purple) performs worst. Unlike in the CNN, the scale advantage of synthetic noise is reduced by LayerNorm (see text).

# H. Centralized Training Results

## H.1. Experimental Details

We evaluate our method across four complexity regimes to demonstrate scalability and modality-agnostic generalization. All methods (DP-SGD, Public DP-KFC, Synthetic DP-KFC) share identical hyperparameters per task to ensure fair comparison. Privacy accounting uses Rényi Differential Privacy (RDP) with $\delta = 1/N$ where $N$ is the training set size.

**Optimizer Selection.** For Transformer architectures (CrossViT, BERT), we use AdamW (Loshchilov & Hutter, 2019) as it consistently outperforms SGD on these models. This makes the DP-SGD baseline equivalent to DP-Adam , a stronger adaptive baseline than vanilla DP-SGD since Adam's diagonal second-moment estimates already capture some local geometry. For CNNs and linear models, we use SGD with momentum.

**(1) Simple Vision – MNIST.** We train a 4-layer CNN from scratch with 2 convolutional layers (16 and 32 filters) followed by 2 fully-connected layers. Training uses 5 epochs, batch size 256, learning rate $10^{-3}$, and SGD with momentum 0.9. Public proxy data: FashionMNIST.

**(2) Complex Vision – CIFAR-100.** We fine-tune CrossViT-Tiny-240 pretrained on ImageNet-1k with the backbone unfrozen for end-to-end training. A 2-layer MLP serves as the classification head. Training uses 5 epochs, batch size 256, learning rate $10^{-3}$, and AdamW optimizer. Public proxy data: CIFAR-10.

**(3) NLP – StackOverflow.** We fine-tune `bert-base-uncased` on next-word prediction, training only the classifier head. Training uses 5 epochs, batch size 64, sequence length 128, learning rate $2\times10^{-4}$, and AdamW optimizer. Public proxy data: AG News.

**(4) NLP – IMDB.** We train logistic regression on binary sentiment classification using TF-IDF features (2000 dimensions). Training uses 20 epochs, batch size 256, learning rate $10^{-1}$, and SGD optimizer. Public proxy data: AG News.

*Table 5.* Test accuracy (%). Underlined: best (within 1 std). Syn.=Synthetic noise, Pub.=Public data preconditioner.

| Dataset | Model | $\epsilon$ | Non-DP | DP-SGD | Syn. KFAC / Pub. KFAC |
|---|---|---|---|---|---|
| StackOverflow | BERT | 0.5 | $\sim$99.0$_{\pm0.5}$ | 78.9$_{\pm7.3}$ | 81.3$_{\pm7.0}$ / $\underline{89.8}_{\pm4.2}$ |
| | | 1 | | 89.5$_{\pm1.8}$ | 91.8$_{\pm1.9}$ / $\underline{96.1}_{\pm1.0}$ |
| | | 2 | | 92.9$_{\pm0.7}$ | 95.4$_{\pm0.3}$ / $\underline{97.5}_{\pm0.3}$ |
| | | 3 | | 93.6$_{\pm0.7}$ | 95.9$_{\pm0.3}$ / $\underline{97.9}_{\pm0.3}$ |
| | | 5 | | 94.2$_{\pm0.7}$ | 96.4$_{\pm0.2}$ / $\underline{98.1}_{\pm0.3}$ |
| | | 8 | | 94.6$_{\pm0.6}$ | 96.5$_{\pm0.1}$ / $\underline{98.2}_{\pm0.2}$ |
| | | 10 | | 94.8$_{\pm0.6}$ | 96.6$_{\pm0.1}$ / $\underline{98.3}_{\pm0.2}$ |
| MNIST | CNN | 0.5 | 98.8$_{\pm0.1}$ | 90.8$_{\pm0.2}$ | 93.3$_{\pm0.6}$ / $\underline{94.7}_{\pm0.4}$ |
| | | 1 | | 91.7$_{\pm0.2}$ | 94.2$_{\pm0.5}$ / $\underline{95.3}_{\pm0.4}$ |
| | | 2 | | 92.5$_{\pm0.3}$ | 95.0$_{\pm0.4}$ / $\underline{95.7}_{\pm0.3}$ |
| | | 3 | | 92.9$_{\pm0.3}$ | 95.3$_{\pm0.4}$ / $\underline{96.0}_{\pm0.3}$ |
| | | 5 | | 93.4$_{\pm0.3}$ | 95.7$_{\pm0.3}$ / $\underline{96.2}_{\pm0.3}$ |
| | | 8 | | 93.7$_{\pm0.3}$ | 95.9$_{\pm0.3}$ / $\underline{96.4}_{\pm0.3}$ |
| | | 10 | | 94.0$_{\pm0.3}$ | 96.0$_{\pm0.3}$ / $\underline{96.5}_{\pm0.2}$ |
| CIFAR-100 | CrossViT | 0.5 | 68.3$_{\pm0.3}$ | 52.1$_{\pm0.5}$ | $\underline{53.6}_{\pm0.5}$ / $\underline{53.1}_{\pm0.5}$ |
| | | 1 | | 56.2$_{\pm0.4}$ | $\underline{57.6}_{\pm0.5}$ / $\underline{57.3}_{\pm0.4}$ |
| | | 2 | | 58.5$_{\pm0.3}$ | $\underline{59.8}_{\pm0.4}$ / $\underline{59.6}_{\pm0.5}$ |
| | | 3 | | 59.5$_{\pm0.3}$ | $\underline{60.9}_{\pm0.3}$ / $\underline{60.6}_{\pm0.4}$ |
| | | 5 | | 60.7$_{\pm0.3}$ | $\underline{62.0}_{\pm0.3}$ / $\underline{61.6}_{\pm0.4}$ |
| | | 8 | | 61.4$_{\pm0.4}$ | $\underline{62.7}_{\pm0.4}$ / $\underline{62.4}_{\pm0.3}$ |
| | | 10 | | 62.0$_{\pm0.4}$ | $\underline{63.2}_{\pm0.3}$ / $\underline{62.9}_{\pm0.3}$ |
| IMDB | LogReg | 0.5 | $\sim$88.0$_{\pm0.5}$ | 82.1$_{\pm0.3}$ | $\underline{83.5}_{\pm0.1}$ / $\underline{83.5}_{\pm0.1}$ |
| | | 1 | | 82.9$_{\pm0.1}$ | $\underline{85.1}_{\pm0.0}$ / $\underline{85.1}_{\pm0.1}$ |
| | | 1.5 | | 83.1$_{\pm0.0}$ | $\underline{85.5}_{\pm0.1}$ / $\underline{85.5}_{\pm0.1}$ |
| | | 2.8 | | 83.1$_{\pm0.1}$ | $\underline{85.9}_{\pm0.1}$ / $\underline{85.8}_{\pm0.1}$ |
| | | 8 | | 83.2$_{\pm0.1}$ | $\underline{86.0}_{\pm0.0}$ / $\underline{86.0}_{\pm0.0}$ |

## H.2. Comparison with Private Optimizer Baselines

Table 1 in the main text compares DP-KFC against representative private optimizers on MNIST (4-layer CNN); Table 6 below gives the same comparison over the full $\epsilon$ range. We group methods by where they act relative to the privacy mechanism. **Pre-privatization (scale-then-privatize, STP)** methods transform the gradient *before* clipping and noise: AdaDPS (Li et al., 2022) (diagonal preconditioner from side information) and our DP-KFC (block-diagonal KFAC preconditioner from synthetic noise). **Post-privatization** methods act on the already-privatized gradient: DP-AdamBC (Tang et al., 2024) debiases Adam's second-moment estimate, and DiSK (Zhang et al., 2025) applies a simplified Kalman filter to denoise the privatized gradient sequence. We use the authors' official implementations, tune learning rate, clipping norm, and method-specific hyperparameters per method via grid search, and report mean $\pm$ std over 5 seeds; privacy accounting is RDP with $\delta = 1/N$ as elsewhere.

*Table 6.* **Comparison with private optimizers on MNIST** (CNN, test accuracy %, 5 seeds, hyperparameters tuned per method). Full $\epsilon$ range; an abridged version is Table 1 in the main text. Top block: standalone optimizers (Public DP-KFC additionally requires a public proxy). Bottom: pre-privatization STP combined with post-privatization bias correction. Bold marks the best entry per column.

| Method | $\epsilon{=}1$ | $\epsilon{=}2$ | $\epsilon{=}5$ | $\epsilon{=}8$ |
|---|---|---|---|---|
| DP-SGD | $91.7_{\pm0.2}$ | $92.5_{\pm0.3}$ | $93.4_{\pm0.3}$ | $93.7_{\pm0.3}$ |
| AdaDPS (Li et al., 2022) | $91.3_{\pm0.8}$ | $93.2_{\pm1.0}$ | $93.6_{\pm1.3}$ | $93.3_{\pm1.4}$ |
| DiSK (Zhang et al., 2025) | $93.7_{\pm0.4}$ | $94.1_{\pm0.3}$ | $94.3_{\pm0.2}$ | $94.3_{\pm0.2}$ |
| DP-AdamBC (Tang et al., 2024) | $94.0_{\pm0.3}$ | $94.8_{\pm0.2}$ | $95.2_{\pm0.2}$ | $95.3_{\pm0.1}$ |
| Public DP-KFC | $95.3_{\pm0.4}$ | $95.7_{\pm0.3}$ | $96.2_{\pm0.3}$ | $96.4_{\pm0.3}$ |
| **Synthetic DP-KFC** | $94.2_{\pm0.5}$ | $95.0_{\pm0.4}$ | $95.7_{\pm0.3}$ | $95.9_{\pm0.3}$ |
| **Synthetic DP-KFC + DP-AdamBC** | $\mathbf{95.5}_{\pm0.3}$ | $\mathbf{96.1}_{\pm0.2}$ | $\mathbf{96.4}_{\pm0.3}$ | $\mathbf{96.4}_{\pm0.3}$ |

Two observations follow. First, among methods that consume *no auxiliary data*, Synthetic DP-KFC attains the best accuracy at every $\epsilon$, and DP-KFC with a (matched) public proxy improves further, consistent with the eigenspectrum analysis of Appendix F, since reshaping the gradient before clipping homogenizes per-parameter sensitivity rather than amplifying privacy noise (cf. Theorem 5.4). Second, because STP and post-privatization corrections target orthogonal failure modes (clipping/conditioning vs. second-moment bias and noise accumulation), they compose: *Synthetic DP-KFC + DP-AdamBC* dominates either method alone across all $\epsilon$. The trend on the main benchmarks points the same way, as DP-KFC already improves over the DP-Adam baseline on the CIFAR-100 (CrossViT) and StackOverflow (BERT) fine-tuning tasks (Table 5), so we expect the standalone ranking and the STP + post-privatization complementarity to persist at larger scale; a full sweep of the post-privatization baselines on the fine-tuning benchmarks is a natural extension.

# I. Hyperparameter Sensitivity & Ablation Study

We conducted Bayesian hyperparameter optimization using Optuna's TPE sampler with 150 trials per method to ensure fair comparison between DP-SGD and DP-KFC. Each trial independently samples learning rate $\eta \in [10^{-3}, 0.5]$, clipping norm $C \in [0.1, 50]$, and for DP-KFC: damping $\pi \in [10^{-5}, 0.1]$, noise exponent $\alpha \in [0, 2]$, and preconditioner steps $\in [1, 100]$.

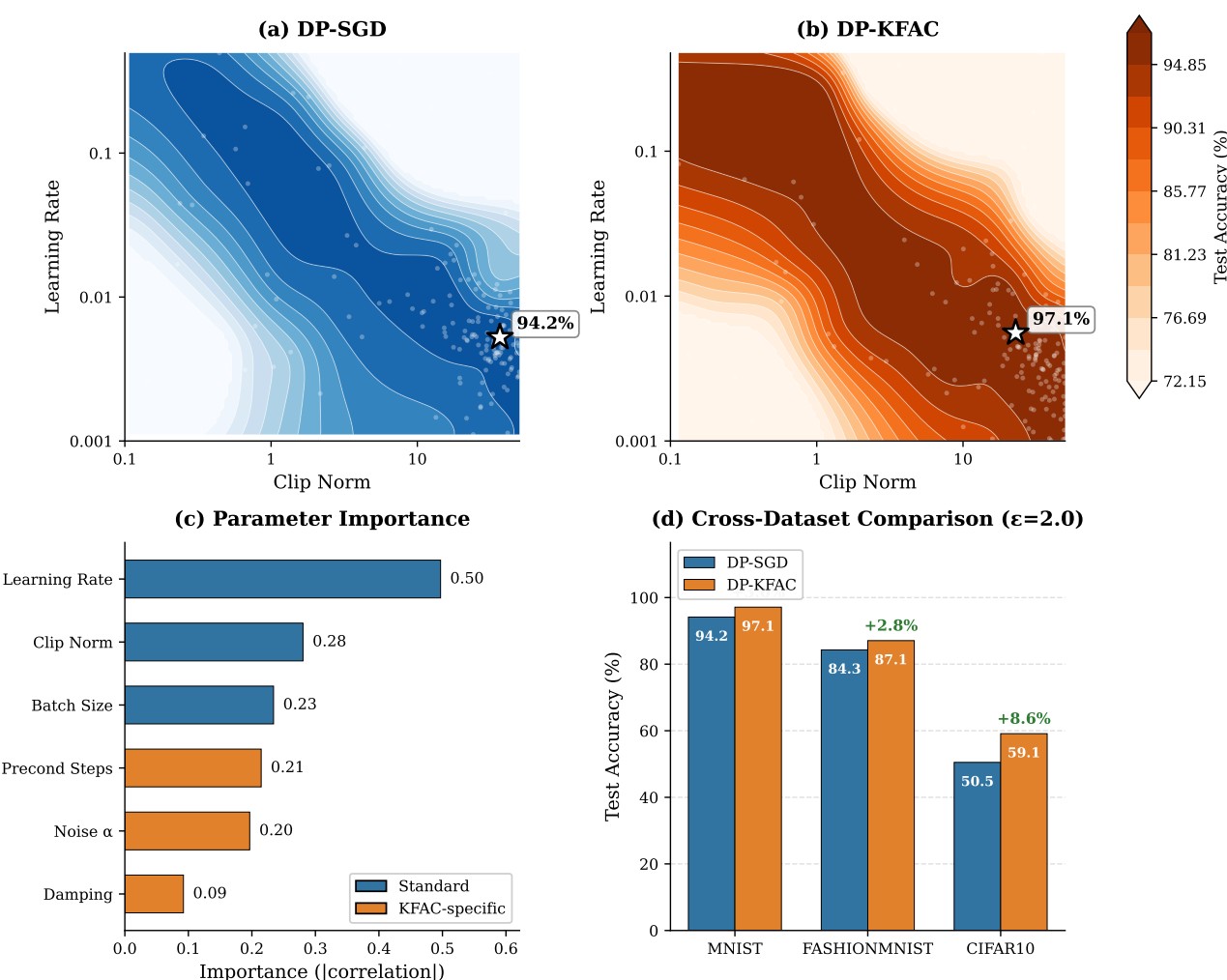

*Figure 10.* **Hyperparameter Optimization Analysis** (MNIST, $\epsilon = 2.0$, 150 Optuna trials per method). **(a, b)** Clip norm $\times$ learning rate interaction surfaces. DP-SGD exhibits a narrow diagonal ridge of trainability, while DP-KFC creates a broad plateau spanning an order of magnitude in clip norm. **(c)** Parameter importance ranking. Standard optimization parameters (blue) dominate; KFAC-specific parameters (orange) have low importance, indicating minimal tuning burden. **(d, e)** DP-KFC parameter interactions show high-accuracy configurations (yellow) across wide ranges of damping and noise $\alpha$. **(f)** Summary: DP-KFC achieves +3.0% improvement with noise $\alpha \approx 1.8$ (pink/brown noise), validating the spectral matching hypothesis.

## I.1. The Decoupling Effect

Panels (a) and (b) of Fig. 10 reveal the central practical advantage of DP-KFC. In standard DP-SGD, the clipping norm $C$ and learning rate $\eta$ are tightly coupled: if $C$ is too small, gradients are highly biased, requiring a large $\eta$ to compensate; if $C$ is too large, the injected noise $\sigma^2 C^2$ explodes, requiring a tiny $\eta$. This creates a narrow diagonal relationship that makes hyperparameter tuning harder.

DP-KFC fundamentally breaks this coupling. The preconditioning step normalizes gradient magnitudes *before* clipping, creating a broad rectangular region of near-optimal performance. As shown in Panel (b), accuracy exceeds 90% across clip norms spanning $[0.5, 50]$, a $100\times$ range, dramatically simplifying hyperparameter search.

## I.2. Minimal Tuning Burden

A common concern with second-order methods is that they introduce additional hyperparameters that require careful tuning. Panel (c) directly addresses this concern by ranking parameter importance based on correlation with final accuracy.

Learning rate dominates (importance = 0.50), followed by clip norm (0.28) and batch size (0.23). This demonstrates that DP-KFC does not introduce additional hyperparameter sensitivity compared to DP-SGD. The method-specific parameters can be set to reasonable defaults ($\pi = 10^{-3}$, $\alpha = 1.0$, steps = 10) with minimal performance degradation.

## I.3. Spectral Validation

Panel (e) provides empirical validation of our theoretical motivation. The optimal noise exponent $\alpha$ consistently clusters around 1.0–2.0 across all datasets, corresponding to pink and brown noise. This confirms that synthetic inputs with $1/f$ spectral decay better approximate natural image statistics, yielding superior curvature estimates compared to white noise ($\alpha = 0$).

## I.4. Cross-Dataset Consistency

Table 7 summarizes the Bayesian optimization results across three vision benchmarks. DP-KFC consistently outperforms DP-SGD, with gains ranging from +2.8% to +8.6%.

*Table 7.* **Bayesian Optimization Results** ($\epsilon = 2.0$, 150 trials per method). DP-KFC consistently outperforms DP-SGD across all datasets, with the largest gains on the most challenging task (CIFAR-10).

| Dataset | DP-SGD | DP-KFC | Δ | Optimal $\alpha$ |
|---|---|---|---|---|
| MNIST | 94.2% | **97.1%** | +2.9% | 1.80 |
| FashionMNIST | 84.3% | **87.1%** | +2.8% | 1.48 |
| CIFAR-10 | 50.5% | **59.1%** | +8.6% | 1.78 |

Notably, the largest improvement occurs on CIFAR-10 (+8.6%), the most challenging dataset with complex natural images. This suggests that DP-KFC's benefits are amplified when the optimization landscape is more difficult, precisely when better curvature information is most valuable.

The optimal noise exponent $\alpha$ ranges from 1.48 to 1.80 across datasets, consistently in the pink-to-brown noise regime. This cross-dataset consistency further validates that matching the $1/f$ spectral statistics of natural images provides a robust architectural prior for curvature estimation.

## I.5. Computational Overhead

The matrix inversions required by DP-KFC introduce a $2.2\times$ slowdown in sample throughput compared to DP-SGD. However, in differential privacy, the optimization horizon is strictly bounded by the privacy budget $\epsilon$, not wall-clock time. Therefore, investing additional computation per step to maximize the utility of each privacy-consuming gradient query is the theoretically optimal strategy.

Moreover, the preconditioner computation is embarrassingly parallel and can be amortized: Panel (c) shows that `precond_steps` has low importance (0.21), and our ablations confirm that 10–20 synthetic batches suffice for robust curvature estimation. When the update frequency is reduced (e.g., recomputing every 5 epochs instead of every epoch), the overhead becomes negligible.

## I.6. Parameter Interaction

To understand how DP-KFC's hyperparameters interact, we visualize pairwise relationships across all 150 Optuna trials (Fig. 11). This analysis reveals which parameters require careful tuning and which can be safely set to defaults.

### I.6.1. THE CLIP-LEARNING RATE COUPLING

Panel (a) reveals that clipping norm and learning rate exhibit a diagonal coupling: larger clip norms require smaller learning rates to maintain optimal performance. This is expected since the effective step size scales with both parameters. However, compared to DP-SGD (Fig. 10a), this diagonal band is substantially *wider*, indicating that DP-KFC's gradient homogenization reduces sensitivity to this interaction.

### I.6.2. DAMPING: A NON-CRITICAL PARAMETER

Panels (b), (d), and (h) consistently show that damping $\pi$ has minimal impact on performance. In panel (b), the yellow region extends vertically across three orders of magnitude ($10^{-4}$ to $10^{-1}$) for any reasonable clip norm. Panel (d) shows horizontal bands indicating that optimal learning rate is independent of damping choice. This contradicts the common belief that second-order methods require careful damping tuning. In practice, $\pi = 10^{-3}$ works well across all our experiments.

### I.6.3. SPECTRAL MATCHING VALIDATION

Panels (c), (e), (g), and (i) provide empirical validation of our spectral matching hypothesis. In each panel, the yellow high-accuracy region concentrates at noise exponent $\alpha > 1.0$, corresponding to pink and brown noise. Panel (c) is particularly striking: regardless of clip norm, $\alpha \approx 1.5$ consistently outperforms white noise ($\alpha = 0$) by 3–5% accuracy. This confirms that synthetic inputs with $1/f$ spectral decay better approximate natural image statistics, yielding superior curvature estimates without access to real data.

### I.6.4. PRECONDITIONER STEPS: MINIMAL OVERHEAD

Panels (f), (h), and (i) demonstrate that the number of preconditioner estimation steps has negligible impact on final accuracy. The horizontal banding pattern in panel (f) shows that accuracy depends only on learning rate, not on whether 10 or 100 synthetic batches were used for curvature estimation. This has important practical implications: the architectural curvature can be reliably estimated with minimal computational overhead, typically saturating after 10–20 batches.

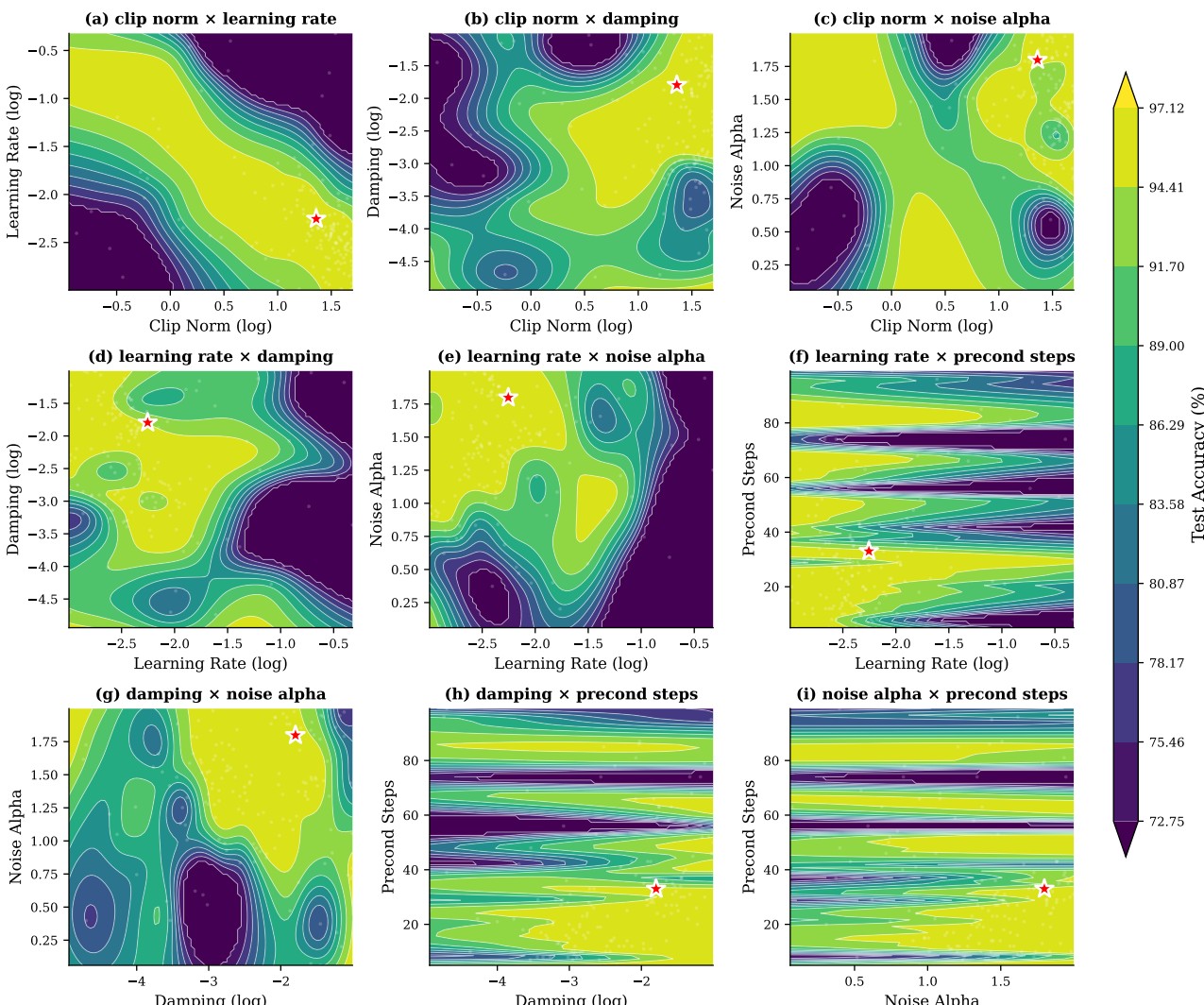

*Figure 11.* **DP-KFC Parameter Interactions** (MNIST, $\epsilon = 2.0$). Each panel shows the accuracy landscape as a function of two hyperparameters. Yellow indicates high accuracy ($> 95\%$), purple indicates low accuracy. Stars mark the best configuration found.

