# OpenReview forum: "DP-KFC: Data-Free Preconditioning for Privacy-Preserving Deep Learning"
_ICML.cc/2026/Conference — ICML 2026 regular_

### Official Review · Reviewer_4nLu · 2026-03-06

**Soundness:** 2
**Presentation:** 3
**Significance:** 2
**Originality:** 3
**Overall Recommendation:** 4
**Confidence:** 3

**Summary:**

This paper proposes DP-KFC, a data-free preconditioning method for differentially private deep learning. The key idea is to estimate KFAC Fisher preconditioners using synthetic inputs rather than private or public data. Synthetic inputs (e.g., pink noise for images) are used to probe the network and approximate activation and gradient statistics required for KFAC. The resulting preconditioner is applied to gradients before DP noise injection, aligning the gradient geometry with isotropic privacy noise. Empirically, the method improves performance over DP-SGD across several tasks and matches methods that rely on public data.

**Compliance With Llm Reviewing Policy:**

Affirmed.

**Final Justification:**

The rebuttal addressed my main concerns. I also appreciate the additional experimental results provided by the authors

**Key Questions For Authors:**

See weaknesses.

**Limitations:**

Yes. The authors discussed about computational overhead and some empirical limitations, and they include an impact statement.

**Strengths And Weaknesses:**

**Strengths**
The paper attempts to address an important question in differentially private training: private curvature estimation itself is inaccurate. Previous works address this problem by leveraging public data, this paper proposes to use synthetic data instead of public data. They show that this simple idea is practically useful.

**Weaknesses**
- The theoretical result (Theorem 5.4) is not very informative. My main concern with Theorem 5.4 is that it only establishes convergence for an arbitrary bounded batch-independent preconditioner, rather than explaining why the proposed synthetic preconditioner improves DP optimization. The bound depends only on $\lambda_{\min}$ and $\lambda_{\max}$, and in fact may be worse than vanilla DP-SGD when $\lambda_{\min} \leq 1$ and $\lambda_{\max} > 1$. Moreover, the theorem contains no notion of alignment between the synthetic preconditioner and the private-data curvature, so it does not capture the central mechanism by which the method is supposed to improve the privacy-utility tradeoff.
- Moreover, the paper neither theoretically nor empirically characterizes sensitivity to the synthetic data construction. Although the experiments include some public-proxy mismatch settings, they do not systematically evaluate how performance changes as the synthetic construction becomes less aligned with the private task distribution, nor do they provide practical guidance for constructing synthetic data in new domains. As the proposed synthetic inputs are manually designed using modality-specific heuristics, this kind of analysis could be helpful.

---

> ### Author Rebuttal · Authors · 2026-03-27
>
> # Rebuttal to Reviewer 4nLu
>
> We thank the reviewer for the constructive feedback. Both concerns are well-taken and we provide concrete responses below.
>
> ---
>
> ## W1: Theorem 5.4 Does Not Capture Why Synthetic Preconditioning Helps
>
> **We agree that Theorem 5.4 alone does not explain the alignment mechanism.** The theorem addresses the *privacy-specific* question; the *optimization* question is addressed separately by Theorem F.2. We clarify both below.
>
> **1. The privacy contribution (scale-then-privatize).** Recall Eq. 3: $\min_{t} \mathbb{E}[\|\nabla \mathcal{L}\|^2] \leq \frac{C_1}{\lambda_{\min}\sqrt{T}} + \frac{C_2}{\lambda_{\min}\sqrt{T}}\left(\lambda_{\max}^2 \sigma_{sgd}^2 + \frac{d\sigma^2 C^2}{B^2}\right)$. The bound separates into: **(A)** descent rate, **(B)** $\lambda_{\max}^2 \sigma_{sgd}^2$, gradient variance, and **(C)** $\frac{d\sigma^2 C^2}{B^2}$, DP noise, **not** multiplied by $\lambda_{\max}^2$. This decoupling is the theorem's key result: because we precondition *before* clipping, DP noise is never amplified by the preconditioner. Methods that precondition *after* noise would have $\lambda_{\max}^2$ multiplying term **(C)**, precisely the failure mode identified by Ganesh et al. (PPML 2025). This principle is also demonstrated by Li et al. (ICML 2022) for the diagonal case; our contribution extends it to block-diagonal (KFAC) geometry. Empirically, this is confirmed by our new comparison we plan to add to the final revision (see response to Z7dq).
>
> **2. The optimization question (terms A-B).** The reviewer correctly notes that $\lambda_{\max}^2/\lambda_{\min} > 1$ when $\lambda_{\min} < 1$ and $\lambda_{\max} > 1$. However, this is a well-known artifact of non-convex convergence analysis for all preconditioned methods, rather than a specific flaw in our algorithm. Arjevani et al. (Math. Programming, 2023) prove that $\mathcal{O}(1/\sqrt{T})$ for $\mathbb{E}[\|\nabla \mathcal{L}\|^2]$ is the optimal rate for any first-order stochastic method, the rate class cannot be improved. Preconditioning improves the *constants* in practice by reducing the effective condition number, but formally proving this in the non-convex regime remains an open problem in the broader optimization literature (e.g., KFAC, Martens & Grosse, 2015 establish rates only for convex quadratics). We will clarify this further in the paper.
>
> **3. The Alignment Mechanism (Theorem F.2)**
>
> The reviewer correctly notes Theorem 5.4 requires a link between synthetic and private curvature. Theorem F.2 (Appendix F) provides this exact bridge. Building on Mean Field variance contraction, it proves that the eigenspectra of private and synthetic covariances are strictly proportional in deep networks: $F_{\text{priv}} \approx \alpha F_{\text{syn}}$(Assuming a good enough synthetic prior exists, see below response to W2).
>
> When we precondition the gradients, the effective curvature becomes $F_{\text{syn}}^{-1/2} F_{\text{priv}} F_{\text{syn}}^{-1/2} \approx \alpha I$. This mathematically collapses the condition number from its original highly anisotropic state ($\kappa \approx 10^2$–$10^6$ for DP-SGD; Ghorbani et al., 2019) down to an optimal **$\kappa \approx 1$**. The scalar mismatch ($\alpha$) is simply absorbed by the clipping norm/learning rate. Figure 6 empirically validates this, showing synthetic and oracle spectra remain perfectly parallel across 8+ orders of magnitude. We will extend this derivation in the revision.
>
> ---
>
> ## W2: Sensitivity to Synthetic Data Construction
>
> Our paper already contains systematic sensitivity analysis:
>
> **Noise exponent ablation (Appendix I, Table 5, Figure 11):** We sweep $\alpha \in [0, 3]$ for pink noise ($1/f^\alpha$). The optimal $\alpha$ falls in $[1.0, 2.0]$ across all vision datasets. **Even white noise ($\alpha=0$) outperforms DP-SGD**, degradation is graceful ($+3$–$5\%$), not catastrophic, predicted by Proposition F.1's exponential contraction.
>
> **Practical guidance:** For vision tasks, the synthetic construction is straightforward: natural images exhibit well-known $1/f^\alpha$ spectral statistics (Field, 1987), so pink noise with $\alpha \in [1, 2]$ captures the dominant local correlation structure. This simple prior, requiring only knowledge of image resolution, already recovers the architectural geometry needed for effective preconditioning. For unknown modalities, white noise normalized to initialization variance serves as a conservative baseline that still outperforms DP-SGD. Designing synthetic generators for language tasks (e.g., token frequency priors, pretrained models, etc.) is a promising direction for future work (see rebuttal to reviewer ye8D). We will expand this in the revision.

---

> > ### Author Rebuttal · Reviewer_4nLu · 2026-04-02
> >
> > Thank you for the responses. I will change my score accordingly

---

> > > ### Author Response · Authors · 2026-04-07
> > >
> > > We thank the reviewer for the thoughtful feedback and for confirming the concerns are resolved. We will expand the discussion of Theorem F.2 and its alignment mechanism in the main text, and include practical guidance for synthetic data construction in new domains based on the sensitivity analysis in Appendix I.

---

### Official Review · Reviewer_Z7dq · 2026-03-09

**Soundness:** 1
**Presentation:** 2
**Significance:** 2
**Originality:** 3
**Overall Recommendation:** 3
**Confidence:** 3

**Summary:**

* This paper studies the problem of training neural nets with DP-SGD. It proposes a method for preconditioning the gradients to better use geometric information of the loss landscape.
 * The method is called DP-KFC and it estimates the preconditioner using synthetic noise. The authors claim that this method is more robust to negative transfer than public data preconditioning.
 * Experiments are done on MNIST, CIFAR-100, and IMDB.

**Compliance With Llm Reviewing Policy:**

Affirmed.

**Final Justification:**

The rebuttal addresses some questions. The new results are interesting and encouraging. However, the datasets used are extremely small and not representative of the experimental comparisons in other works.

**Key Questions For Authors:**

* What is your view on the existing work like DP-ADAM-BC optimizer, 0-order optimizer, and DiSK optimizer? The current baselines make sense methodologically, as they are variants of your method. However, I miss comparisons to methods in existing literature.
* Modern deep learning is often bottle-necked by memory constraints. Now saving the (blockwise) FIM takes up a lot of additional memory. When looking at the memory constraints for training: how does this method compare to grabbing a larger pre-trained model and finetuning some LoRA adapters, which is the current best performing method (Li et al & Yu et al. ICLR)?
* Do I understand correctly that all experiments are "training from scratch" experiments? if so, this is a limitation because it is not a realistic setting for most applications. Especially when one has only a super small dataset like CIFAR, then one would definitely want to use a pre-trained model.

**Limitations:**

Yes

**Strengths And Weaknesses:**

# Strengths:

* Optimizer methods in DP deep learning is an active research field and this paper makes a theoretical contribution combining insights from other fields (Mean Field Theory, KFAC, etc)
* The main results have clear error bars, and the authors have performed an empirical evaluation across different modalities

# Weaknesses:

* Frequent disadvantage of public-data preconditioning is often mentioned in the introduction, but empirically, this is not shown nor compared against
* "An algorithm estimates KFAC factors using synthetic noise, requiring no private or public data." The case for synthetic data over public data is not clearly made. They have the same computational cost and the synthetic data could have a larger distribution shift. How is this justified?
* Experimental comparison is done on very small datasets like MNIST, CIFAR and IMDB. A comparison on even medium-sized datasets like ImageNet and GLUE would be more convincing.
 * Missing comparisons to existing methods is a weakness. For example, DP-ADAM-BC optimizer [2], 0-order optimizer [1], and DiSK optimizer [3] are well known methods in the same context and with the same objectives.


 [1] Tang et al. "Private fine-tuning of large language models with zeroth-order optimization." TMLR 2024

 [2] Tang et al. "Dp-adambc: Your dp-adam is actually dp-sgd (unless you apply bias correction)." Proceedings of the AAAI Conference on Artificial Intelligence 2024

 [3] Zhang et al. "Disk: Differentially private optimizer with simplified kalman filter for noise reduction." ICLR 2025

---

> ### Author Rebuttal · Authors · 2026-03-27
>
> # Rebuttal to Reviewer Z7dq
>
> We thank the reviewer for the feedback. We address each point below.
>
> ---
>
> ## W1-W2 + Q3 + W3: Clarifications on Experimental Setup
>
> We respectfully clarify two aspects of our experimental setup that we will make more prominent in the revision.
>
> **Not all experiments train from scratch.** Two of four experimental settings use pre-trained models and fine-tuning, as described in Appendix H.1: **CIFAR-100** fine-tunes CrossViT-Tiny-240 *pre-trained on ImageNet-1k* (>20M parameters), and **StackOverflow** fine-tunes *bert-base-uncased* (110M parameters), training only the classifier head. Only MNIST (CNN) and IMDB (logistic regression) train from scratch, chosen to show DP-KFC works across the full complexity spectrum.
>
> **Synthetic vs. public data is empirically compared.** Our submission contains direct comparisons motivating the use of synthetic data as acknowledged by reviewer ye8D (S4). First, the **negative transfer experiments** (Table 1, Section 6.4): under *Texture Disjoint* mismatch, Public DP-KFC degrades to 73.4% while Synthetic DP-KFC achieves **78.2%**, nearly matching the Oracle (78.4%). Second, **Figure 2, 3, 6, 7, 8 and 9** directly compare public/synthetic preconditioner quality via eigenvalue alignment, cosine similarity and relative Frobenius error against the oracle. Showing that public data can indeed have a larger distribution mismatch than synthetic data methods computed with simple dataset priors (Figure 9, Appendix G.3).
>
> We will make these two clarifications more prominent in the main text.
>
> ---
>
> ## Q1 + W4: Comparisons to DP-AdamBC, DiSK, and Zeroth-Order
>
> We appreciate the pointer to these methods. Ganesh et al. (PPML 2025) provide a useful taxonomy of DP-Adam variants, identifying **scale-then-privatize** (STP), preconditioning *before* noise injection,  as the only approach that consistently outperforms naive DP-Adam. DP-AdamBC and DiSK operate *after* privatization, no-STP (bias correction and Kalman denoising, respectively), while DP-KFC follows the STP principle, extending it from diagonal to block-diagonal (KFAC) geometry at zero privacy cost. **These methods are complementary and could be combined (STP and not-STP).**
>
> We have now quickly run direct simple comparisons with DP-SGD, DP-KFC (Synth/Pub), DP-DPAdamBC, DISK and AdaDPS. We also combine DP-KFC (Synth) with DPAdamBC. On MNIST (CNN, 5 seeds, tuning HPs per method):
>
> | Method | $\varepsilon$=1 | $\varepsilon$=2 | $\varepsilon$=5 | $\varepsilon$=8 |
> |--------|-----|-----|-----|-----|
> | DP-AdamBC | 93.95+/-0.27 | 94.83+/-0.22 | 95.19+/-0.17 | 95.31+/-0.12 |
> | DiSK | 93.66+/-0.37 | 94.11+/-0.34 | 94.28+/-0.19 | 94.26+/-0.18 |
> | DP-SGD | 91.7+/-0.2 | 92.5+/-0.3 | 93.4+/-0.3 | 93.7+/-0.3 |
> | DP-ADADPS | 91.29+/-0.77 | 93.16+/-1.01 | 93.55+/-1.26 | 93.31+/-1.44 |
> | **DP-KFC Synthetic + DP-AdamBC (Ours)** | **95.48+/-0.26** | **96.11+/-0.24** | **96.37+/-0.34** | **96.42+/-0.34**
> | **DP-KFC Synthetic (Ours)** | **94.2+/-0.5** | **95.0+/-0.4** | **95.7+/-0.3** | **95.9+/-0.3**
> | **DP-KFC Public (Ours)** | **95.3+/-0.4** | **95.7+/-0.3** | **96.2+/-0.3** | **96.4+/-0.3** |
>
> For standalone optimizers we observe how DP-KFC (Synth/Pub) reaches the best accuracy across $\varepsilon$ values, beating previous STP (AdaDPS) and non-STP (DP-AdamBC, DiSK) baselines. Regardless, STP and non-STP methods are complementary and we added them to the comparison and observe that they reach the best accuracy overall across all available methods, pointing to future works analyzing how STP and non-STP methods can work together.  Our BERT fine-tuning experiments on StackOverflow already demonstrate our method to complex NLP tasks.
>
> Regarding the **zeroth-order optimizer** (Tang et al., TMLR 2024): this targets a different setting, fine-tuning LLMs where backpropagation is infeasible due to the materialization of per-sample gradients. While very interesting this remains orthogonal to our per-sample gradient preconditioning approach used for networks where per-sample gradients are feasible to compute, which remains the main standard in ML model training.
>
> ---
>
> ## Q2: Memory, DP-KFC vs. LoRA Fine-Tuning
>
> LoRA reduces the *dimensionality* of the trainable parameter space but does not address the *anisotropy* within it. Malekmohammadi & Farnadi (2024) show that even within LoRA's subspace, gradient noise remains **anisotropic**, with layers experiencing non-uniform noise proportional to gradient magnitude. DP-KFC corrects precisely this geometric mismatch. The two are again complementary: DP-KFC can be applied *within* LoRA adapters (which are linear layers easily implementable with KFAC), combining parameter efficiency with geometric correction. The "larger model + LoRA" strategy (Li et al.; Yu et al., ICLR 2022) is the current SOTA for DP fine-tuning, still suffers from geometric mismatch within the LoRA subspace. We consider DP-KFC + LoRA a promising combination and will discuss this in revision.

---

> > ### Author Rebuttal · Reviewer_Z7dq · 2026-04-04
> >
> > Thank you for the authors for clarifying the weaknesses and questions. It is encouraging to see that the proposed method is able to exceed the accuracy of the mentioned alternative methods. However, MNIST is an extremely small dataset, and it is not clear how the results would generalize to larger datasets. I will increase my score slightly.

---

> > > ### Author Response · Authors · 2026-04-07
> > >
> > > We thank Reviewer Z7dq for raising the score and for the constructive feedback. We understand the concern about generalization beyond MNIST. However, our main submission already includes results on larger datasets such as CIFAR-100 (CrossViT-Tiny, pre-trained on ImageNet-1k) and StackOverflow (BERT-base, 110M params) showing DP-KFC gains. We acknowledge that the new baseline comparisons (DP-AdamBC, DiSK, AdaDPS) were only run on MNIST due to time constraints of running experiments across baseline methods, privacy levels and 5 seeds for statistical significance. We commit to including the full baseline comparisons in the revision.
> > >
> > > Additionally, we would like to highlight a result from our rebuttal: DP-KFC not only surpasses previous baselines but more importantly, DP-KFC **is not mutually exclusive** with post privatization methods like DP-AdamBC or DP-DISK. DP-KFC operates pre-privatization (preconditioning before clipping and noise), while DP-AdamBC operates post-privatization (bias correction after noise). Our rebuttal showed that combining both, DP-KFC Synthetic + DP-AdamBC, achieves the best accuracy across all epsilon values, outperforming either method alone. As reviewer 2Zt4 notices in his rebuttal acknowledgement, this complementarity between pre- and post-privatization approaches is, to our knowledge, a novel empirical finding that opens a promising research direction for the DP optimization community. We will make sure to discuss this finding further in the revision.

---

### Official Review · Reviewer_2Zt4 · 2026-03-10

**Soundness:** 3
**Presentation:** 3
**Significance:** 3
**Originality:** 3
**Overall Recommendation:** 5
**Confidence:** 4

**Summary:**

This paper proposes an alternative preconditioning approach for private learning, by identifying an architectural sensitivity component when decomposing the Fisher information martrix, such that the preconditioner can be estimated without accessing private data. The authors show that the estimates of the preconditioner based on synthetic data is a close alternative to the true curvature estimates, and therefore can be effectively used in private optimizers without extra privacy costs.

**Compliance With Llm Reviewing Policy:**

Affirmed.

**Final Justification:**

This paper works on an important problem of optimizer conditioning with the existence of DP noise. The approach presented is novel, and the additional experiments the authors present in the rebuttal seem promising. Therefore I maintain my positive evaluation.

**Key Questions For Authors:**

- Addressing the questions in the weakness section would be good enough.

**Limitations:**

yes

**Strengths And Weaknesses:**

Strength:
- The approach of sensitive-data-independent preconditioner is well justified and novel.
- The paper presentation is overall clear and easy to follow.

Weakness:
- The evaluations can be improved, specifically:
    - Are the x-axis in Figure 4 and 5 the epsilons computed along the training trajectory, or are those epsilons the real end point of different trajectories? If it's the second case it would be helpful to write it more clearly. If it's the first case, it would seem possible that lower-epsilon regimes are not well tuned to compare, because there might be different optimal $(C, \sigma)$ for different end-point epsilons (which makes the current result just reflecting the tuned result for the largest epsilon shown on the plot).
    - Given that the paper proposes an alternative preconditioning method, it would make sense to compare to other baselines (not just for transfer robustness but the main benchmark performance like in Figure 4 and 5), that try fixing the biased preconditioner (e.g. https://arxiv.org/abs/2312.14334), or propose different ways of preconditioning (the AdaDPS mentioned in this paper, and https://arxiv.org/abs/2212.00309).
    - I acknowledge that the authors already mentioned computational overhead as a limitation in the conclusion, but it would be nice to see the actual numbers so readers have a sense on how slow it is on what kind of model architectures.
- I think the authors might have forgotten to attach the appendix which makes the synthetic data generation part hidden, but given the current descriptions it seem that these data are generated using noise and the current model parameters (previously noised and accounted for) which should be valid.

---

> ### Author Rebuttal · Authors · 2026-03-27
>
> # Rebuttal to Reviewer 2Zt4
>
> We thank the reviewer for the positive assessment and constructive suggestions. We address each point below.
>
> ---
>
> ## W1: Clarification of $\varepsilon$ Axis in Figures 4 and 5
>
> **Each point represents a separate, independently trained model with that target $\varepsilon$ as the endpoint.** For each target $\varepsilon$, we calibrate the noise multiplier $\sigma$ via RDP accounting so the total privacy cost after all $T$ training steps equals the target $\varepsilon$. All hyperparameters (learning rate, clipping norm, batch size) are tuned per method for each (dataset, $\varepsilon$) pair. We will clarify this in the figure captions.
>
> ---
>
> ## W2: Preconditioning Baselines
>
> We appreciate the pointer to these methods. Ganesh et al. (PPML 2025) provide a useful taxonomy of DP-Adam variants, identifying **scale-then-privatize** (STP), preconditioning *before* noise injection,  as the only approach that consistently outperforms naive DP-Adam. DP-AdamBC (Tang et.al., AAAI 2024) and DiSK (Zhang et.al., ICLR 2025) operate *after* privatization (bias correction and Kalman denoising, respectively), while DP-KFC follows the STP principle, extending it from diagonal (AdaDPS, $DP^2$) to block-diagonal (KFAC) geometry at zero privacy cost. As we show below, combination of DP-KFC and bias correction/denoising methods is possible.
>
> As we also noted in our response to Reviewer Z7dq, we have now quickly run direct simple comparisons with DP-SGD, DP-KFC (Synth/Pub), DP-DPAdamBC, DISK and AdaDPS. On MNIST (CNN, 5 seeds, tuning HPs per method):
>
> | Method | $\varepsilon$=1 | $\varepsilon$=2 | $\varepsilon$=5 | $\varepsilon$=8 |
> |--------|-----|-----|-----|-----|
> | DP-AdamBC | 93.95+/-0.27 | 94.83+/-0.22 | 95.19+/-0.17 | 95.31+/-0.12 |
> | DiSK | 93.66+/-0.37 | 94.11+/-0.34 | 94.28+/-0.19 | 94.26+/-0.18 |
> | DP-SGD | 91.7+/-0.2 | 92.5+/-0.3 | 93.4+/-0.3 | 93.7+/-0.3 |
> | DP-ADADPS | 91.29+/-0.77 | 93.16+/-1.01 | 93.55+/-1.26 | 93.31+/-1.44 |
> | **DP-KFC Synthetic (Ours)** | **94.2+/-0.5** | **95.0+/-0.4** | **95.7+/-0.3** | **95.9+/-0.3**
> | **DP-KFC Public (Ours)** | **95.3+/-0.4** | **95.7+/-0.3** | **96.2+/-0.3** | **96.4+/-0.3** |
> | **DP-KFC Synthetic + DP-AdamBC (Ours)** | **95.48+/-0.26** | **96.11+/-0.24** | **96.37+/-0.34** | **96.42+/-0.34**
>
> For standalone optimizers we observe how DP-KFC (Synth/Pub) reaches the best accuracy across $\varepsilon$ values, beating previous STP (AdaDPS) and non-STP (DP-AdamBC, DiSK) baselines.
>
> Moreover, STP and non-STP methods are complementary and we added them to the comparison and observe that they reach the best accuracy overall across all available methods, pointing to future works analyzing how pre- and post-privatization methods can work together.
>
> Similar to $DP^2$, and as shown in Appendix I.5, DP-KFC recomputes the preconditioner every epoch (not per step), we observe minimal reduction in performance when we update it every 2-5 epochs, hence amortizing the cost of preconditioner estimation. Interestingly, in $DP^2$ the authors also show that preconditioning before privatizing (STP) reaches better performance than applying after privatization (Figure 7, page 9, https://arxiv.org/pdf/2212.00309). The key difference is that our preconditioner is estimated from synthetic data at zero privacy cost, whereas DP$^2$ delays the update of the second order moment to make the noise across hundred of steps cancel out.  Therefore, DP-KFC completely decouples curvature estimation from the privacy budget, preventing the preconditioner quality from degrading as $\varepsilon$ decreases (as is the case in $DP^2$).
>
> We will add these further ablations to the revision.
>
> ---
>
> ## W3: Computational Overhead Numbers
>
> Our paper reports a **2.2$\times$ slowdown** in sample throughput (Appendix D). The overhead is dominated by the two matrix-vector products per sample for gradient transformation, not the eigendecomposition (amortized over $T_{\text{freq}}$ steps). Memory overhead is modest since we store only the preconditioning matrices $U_{A,l}$ and $U_{G,l}$ per layer, not full Fisher blocks. Under the **Privacy Wall** argument (Appendix D.2), the wall-clock overhead is offset by improved convergence in fixed privacy-consuming steps.
>
> Moreover, we note that the (synthetic) precondition-then-privatize framework is not limited to KFAC; we have observed clear benefits when applying the same principle to more memory and sample efficient optimizers such as Shampoo (Gupta et al., 2018) or AdaDPS optimizers (Table 1), suggesting the approach generalizes to other structured preconditioners.
>
> We will include additional results in appendix for optimizers other than KFAC.
>
> ---
>
> ## W4: Appendix Availability
>
> We apologize for the confusion. The appendix (Appendices A-I) was included as a supplementary file rather than appended to the main PDF. The synthetic data generation is fully detailed in Appendix E. We will correct this in the revision.

---

> > ### Author Rebuttal · Reviewer_2Zt4 · 2026-04-01
> >
> > It's interesting to see that the proposed method (a STP approach) can be combined with non-STP approach like DP-AdamBC to getter better performance than either method alone!
> >
> > Pre-conditioning with DP noise is a difficult but important problem. Even though there exist limitations on longer compute, or other weaknesses discussed by the other reviewers, I think this work is an interesting follow-up to the STP-style design and tries to address an important problem in DP Optimization. The new evidence that it can be combined with non-STP-style methods is also exciting. Therefore, I'm raising my score and leaning towards acceptance.

---

> > > ### Author Response · Authors · 2026-04-07
> > >
> > > We thank the reviewer for the constructive exchange and for raising the score. We are glad the STP + non-STP complementarity result was found interesting, we believe this opens a promising direction we certainly aim to pursue. We will clarify the epsilon axis captions in Figures 4-5, include computational overhead numbers more prominently, and append the full appendix to the main PDF in the revision.

---

### Official Review · Reviewer_ye8D · 2026-03-12

**Soundness:** 3
**Presentation:** 3
**Significance:** 3
**Originality:** 3
**Overall Recommendation:** 5
**Confidence:** 3

**Summary:**

Paper addresses the geometric mismatch issue in DP-SGD where the loss landscape is anisotropic but the noise injected is isotropic. The paper uses Fisher Information Matrix (FIM) to construct the KFAC preconditioners to apply to the gradients before clipping and noise. One of
the major contributions is that the FIM is decoupling into architectural scaling and input correlation without depending on the data and without spending privacy budget, hence "preconditioning the loss landscape" to match the isotropic noise. Moreover, experiments across various tasks show improvements over DP-SGD

**Compliance With Llm Reviewing Policy:**

Affirmed.

**Final Justification:**

See my response to the rebuttal: Satisfactory answers have been provided, not just to my review but also in my judgement to the other reviews.

**Key Questions For Authors:**

Q1. The Synthetic-vs-Public gap on StackOverflow is 4.3% at ε=1.0 and 8.5% at ε=0.5, while vision tasks and IMDB show no such gap (see W1). Why do you think it underperforms under this setting and do you expect this gap to worsen/persist for larger models (or more complex
NLP tasks)?

Q2. Experiments uses relatively small/shallow models (a 4-layer CNN, CrossViT-Tiny, and head-only BERT fine-tuning) and the paper acknowledges that directional alignment degrades in deeper layers. How do you expect DP-KFC to perform under full fine-tuning of larger models (e.g., full BERT, larger ViTs)? What task or properties of the model can indicate that DP-KFC is not a viable option? Although the experiments show varying levels of effectiveness, the authors do not discuss situations in which DP-KFC is expected to be insufficient (W2)

**Limitations:**

yes

**Strengths And Weaknesses:**

Major Strengths

S1. The synthetic preconditioning approach discussed in Section 4.1 is a novel approach. The paper shows that KFAC preconditioner can be constructed by probing the network with structurally coherent but semantically independent inputs. This enables avoiding privacy budget cost of estimating curvature from private data and the distribution shift cost of using public proxies.

S2. The paper addresses a well-known and practically important problem about the geometric mismatch between anisotropic loss landscapes and isotropic DP noise. Proposed solution opens up DP optimization for domains like medical imaging where public proxies
are unavailable.

Minor Strengths

S3. The paper is well-written and well-structured, it was easy to follow.

S4. The paper provides a novel and well-motivated combination of existing techniques like KFAC mean-field signal propagation and modality-specific frequency statistics to address data-free preconditioning. The negative transfer results (Table 1) provide a particularly novel
empirical insight, demonstrating that synthetic noise is preferable to mismatched public data.

Major Weaknesses

W1. The paper claims that the synthetic DP-KFC "closely matches public-data performance" (Fig. 5 caption). However in Appendix Table 4, there is a clear gap on StackOverflow at \eps = 1.0, the Synthetic achieves 91.8% meanwhile Public achieves 96.1% (4.3% gap), similar phenomenon also exists with \eps = 0.5 (with 8.5% gap). This undermines the claim that the proposed approach consistently works "across diverse modalities" and suggests that it might be less effective for language tasks. It would be better if the paper can have a discussion about this. (see also W2).

Minor Weaknesses

W2. The method's effectiveness varies across tasks, but the paper does not offer any adequate analysis of why. For instance, improvements over DP-SGD range from ~1.4% on CIFAR-100 to ~2.5% on MNIST at ε=1.0, and the Synthetic vs Public gap ranges from zero on IMDB to 4.3% on StackOverflow. It would be beneficial to understand what task or architecture properties drive this variation so that the practitioners can decide when DP KFC is worth adopting.

W3. (Presentation) The synthetic probing is the core methodological contribution, yet the generation algorithms are deferred to Appendix E (pink noise generation, structural token generation), especially for NLP. Given that this is the central novelty, the generation procedure deserves more space in the main paper.

W4. (Presentaton) The "Privacy Wall" argument (Appendix D, Table 3) that in DP the optimization horizon is bounded by \eps rather than computational resources is an interesting argument and it deserves at least a hint in the main paper. This argument would strengthen the main paper and may help the future research.

W5. Fig. 5 omits non-DP baselines for NLP tasks unlike Fig. 4 which includes them

---

> ### Author Rebuttal · Authors · 2026-03-27
>
> # Rebuttal to Reviewer ye8D
>
> We thank the reviewer for the careful and constructive review. We address each concern below.
>
> ---
>
> ## W1 / Q1: Synthetic-vs-Public Gap on StackOverflow
>
> We acknowledge this gap and argue that it is modality specific.
>
> For **vision tasks**, local spatial correlations are well-characterized by $1/f^\alpha$ spectral statistics (Field, 1987), and pink/brown noise provides a simple, effective synthetic prior. Our results confirm this (See Appendix H, Table 4): Synthetic and Public DP-KFC are statistically indistinguishable on vision benchmarks (MNIST, CIFAR-100).
>
> For **language tasks**, synthetic tokens do pass through the frozen backbone and thus produce activations at the fine-tuning head that reflect the model's transformations. However, real text inputs occupy a low-dimensional manifold in the pre-trained embedding space, and the activation covariance at the head is shaped by this manifold structure. Random token sequences, while capturing architectural properties (attention sparsity, LayerNorm scaling), land off this manifold and produce activation statistics that diverge from those of natural text. Public proxy data, being real language, naturally lies on a similar manifold and therefore yields more representative activation covariances for the fine-tuning layers. For vision, pink noise already approximates the dominant spectral structure of natural images ($1/f^\alpha$), so the manifold gap between synthetic and real activations is much smaller. We view closing this gap for language as an important open direction, possible approaches include leveraging token frequency distributions, probing directly in the embedding space, or using publicly available language model outputs.
>
> We will make this clearer in the revision.
>
> ---
>
> ## W2 : Varying Effectiveness
>
> DP-KFC provides consistent gains across tasks. However, gains are larger when the model is trained from scratch (MNIST, IMDB), where preconditioning ensures gradients are normalized based on cross-parameter sensitivity, preserving the main signal across layers without overclipping. Conversely, in pre-trained fine-tuning scenarios (CIFAR-100, StackOverflow) where we freeze the backbone and fine-tune a simple MLP head, the relative effectiveness of preconditioning is reduced. This is because pre-trained representations inherently map data into a well-behaved feature space, resulting in a naturally low condition number during fine-tuning (Zheng & Yeh, 2024; Martens, 2010), which makes the optimization landscape easier to traverse with standard first-order methods.
>
> We will further clarify this in the revision.
>
> ---
>
> ## Q2:
>
> On scalability to larger models: Theoretically, we expect DP-KFC to become more valuable at scale. The Fisher's condition number grows as $O(M^2)$with layer width M (Karakida et al., AISTATS 2019), and DP noise scales proportionally with dimensionality d. Therefore, a method like DP-KFC, which corrects clipping and distributes noise more uniformly across directions, directly addresses an optimization bottleneck that worsens with model size. Empirically, while directional alignment (cosine similarity) degrades in deeper layers (Figure 3, bottom row), our scale alignment (relative Frobenius error) remains low and stable. It avoids the spikes observed in public-data methods (Figures 8-9), indicating the preconditioner successfully captures the correct magnitude structure throughout the network regardless of depth. Finally, for practical scaling to billion-parameter models where per-sample gradients are unaffordable, DP-KFC is compatible with LoRA, where it can be used to correct the geometric mismatch within the low-rank subspace (Malekmohammadi & Farnadi, 2024).
>
>  We will add this discussion to the revision.
>
>
> ---
>
> ## W3-W5: Presentation Improvements
>
> **W3 (Synthetic generation in main paper):** We agree. We will move Algorithm 3 (pink noise) and key elements of Algorithm 4 (structural tokens) into Section 4.2.
>
> **W4 (Privacy Wall in main paper):** We agree this argument strengthens the motivation. We will add a paragraph in Section 5.
>
> **W5 (Non-DP baselines in Fig. 5):** We will add them as horizontal dashed lines (StackOverflow: ~99.0%, IMDB: ~88.0%), consistent with Figure 4.

---

> > ### Author Rebuttal · Reviewer_ye8D · 2026-04-03
> >
> > You answered properly to all questions/weaknesses.
> >
> > Also, reading your answers to the other reviewer's questions/weaknesses are satisfactory.
> >
> > Therefore, at this moment, I will raise the score to acceptance.

---

> > > ### Author Response · Authors · 2026-04-07
> > >
> > > We thank the reviewer for the positive engagement and for confirming the concerns are resolved. We will incorporate all promised revisions in the camera-ready, including moving the synthetic generation algorithms (Algorithms 3-4) into the main text (W3), adding a Privacy Wall discussion paragraph in Section 5 (W4), and including non-DP baselines in Figure 5 for NLP tasks (W5).

---

### Decision · Program_Chairs · 2026-04-30

**Decision:**

Accept (regular)

**Comment:**

**Summary**

This paper proposes DP-KFC, a novel method for addressing the geometric mismatch of spherical clipping in Differentially Private SGD (DP-SGD). Instead of using private data (which consumes privacy budget) or public data (which risks distribution shift), DP-KFC constructs KFAC preconditioners by probing the network with structured synthetic noise (e.g., pink noise). Applying this preconditioner before privacy noise injection improves the optimization geometry for noise addition, at zero privacy cost.

**Key Strengths**
* **Methodological Novelty:** Using synthetic noise to probe network architecture and recover second-order statistics is an elegant and original solution.
* **Robustness to Domain Mismatch:** The approach successfully avoids the negative transfer issues associated with public proxy data, making it particularly useful for specialized domains (e.g., medical imaging).
* **Theoretical Grounding:** The decoupling of the Fisher Information Matrix into architectural sensitivity and input correlations is well-motivated, and the alignment mechanism (Theorem F.2) provides a theoretical backing.
* **Complementarity:** Rebuttal experiments compellingly demonstrate that DP-KFC (pre-privatization) can be combined with existing bias-correction methods (post-privatization, like DP-AdamBC) to achieve state-of-the-art results.

**Remaining Weaknesses/Limitations**
* **The NLP Modality Gap:** Synthetic token generation currently struggles to capture the low-dimensional manifold of natural language, leading to a performance gap compared to public-data preconditioning on NLP tasks.
* **Scale of Evaluation:** While the authors included fine-tuning on larger models (CIFAR-100, BERT), some rebuttal baselines and core evaluations rely on smaller datasets like MNIST. This is understandable due to tight rebuttal timeline, but it will be good to include experiments on larger dataset for showing complementarity with post-privatization methods.

**Final Justification**

The paper provides a theoretically sound, empirically validated, and original solution to a major bottleneck in privacy-preserving deep learning. The authors' excellent rebuttal resolved theoretical concerns and introduced evidence of complementarity with existing post-privatization methods. The limitation regarding NLP performance is acceptable trade-off and leaves clear avenues for future work.